# Evolution of sexual size dimorphism in tetrapods is driven by varying patterns of sex-specific selection on size

**Alex Slavenko** [1] ✉, **Natalie Cooper** [2], **Shai Meiri**[3,4], **Gopal Murali** [5], **Daniel Pincheira-Donoso**[6] & **Gavin H. Thomas** [7] ✉

Sexual size dimorphism (SSD) is highly prevalent in nature. Several hypotheses aim to explain its evolution including sexual selection, differential equilibrium and ecological niche divergence. Disentangling the causal mechanism behind the evolution of SSD is challenging, as selection arising from multiple pressures on fitness may act simultaneously to generate observed patterns. Here, we use phylogenetic comparative methods to study the evolution of SSD across tetrapods globally. We estimate directional changes in body size evolution, and compare the number, phylogenetic position and magnitude of size changes between sexes. We find evidence that directional changes in size associated with SSD are typically more common in males—even in lineages where females are larger. However, underlying mechanisms differ among lineages—whereas SSD in amphibians becomes more male-biased with greater increases in male size and mammalian SSD becomes more female-biased with greater decreases in male size. Thus, differing mechanisms of directional body size evolution across sexes are essential to explain observed SSD patterns.

Selection on animal body size is often sex-specific, leading to the evolution of sexual size dimorphism (SSD)—the difference in size between males and females of the same species[1,2]. SSD is widespread across the animal tree of life and varies extensively both in magnitude and direction[1–11]. SSD can be either female- or male-biased (females or males are the larger sex, respectively), whereas in other species both sexes have similar sizes (sexual monomorphism). The strength of sex-specific selection varies immensely across species, thus shaping a vast diversity of levels of SSD, from species where sexes barely differ in size (for example, in humans[12]), to species where one sex is extremely large relative to the other (for example, in web-building spiders of the genus *Argiope*[13] where females can be up to five times the length of males or in the cichlid fish *Lamprologus callipterus* where males are over 12 times heavier than females[14]).

Such striking diversity in SSD across lineages has led to the formulation of a range of hypotheses that invoke mechanisms as diverse as competition over access to mates, differences in sex roles (for example, fecundity and sexual conflict) and divergent natural selection that drives intersexual niche divergence to mitigate ecological competition between the sexes[1–3,15,16]. Some of the earliest, and most common, hypotheses to explain the evolution of SSD relate to sexual selection[15]. When access to female mates is enhanced by larger male size (for example, under male–male competition or female preference for larger males), sexual selection is predicted to lead to a size increase in males relative to females, resulting in male-biased SSD[15,17,18]. In contrast, female-biased SSD is predicted when female-specific fitness correlates positively with brood size, resulting in fecundity selection for larger females relative to males[15,16,19]. These selection mechanisms are not

[1]Cesar Australia, Brunswick, Victoria, Australia. [2]Natural History Museum London, London, UK. [3]School of Zoology, Tel Aviv University, Tel Aviv, Israel. [4]The Steinhardt Museum of Natural History, Tel Aviv, Israel. [5]Department of Ecology and Evolutionary Biology, University of Arizona, Tuscson, AZ, USA. [6]MacroBiodiversity Lab, School of Biological Sciences, Queen's University Belfast, Belfast, UK. [7]School of Biosciences, University of Sheffield, Sheffield, UK. ✉e-mail: alex.slavenko1@gmail.com; gavin.thomas@sheffield.ac.uk

mutually exclusive—they can act on both sexes simultaneously, and if optimal sizes differ between the sexes SSD can evolve (the 'differential equilibrium' model[20]). Selection for smaller size of one sex can also lead to SSD. For example, sexual selection can lead to female-biased SSD if smaller males are selected for, as in acrobatic aerial displays in shorebirds[21]. Likewise, selection for small size in female weasels (*Mustela* spp.) may allow pregnant females to enter prey burrows, and has been advocated as a cause of their male-biased SSD[22].

More recently, intersexual divergent natural selection has been invoked as a potentially widespread source of (ecological) sexual dimorphism. This hypothesis predicts that, in populations where ecological competition intensifies between males and females that overlap in the use of similar resource sets, sexual conflicts arise between them[1,15,23–26]. In these situations, intersexual ecological conflict is predicted to be mitigated via the emergence of sex-specific natural selection that drives females and males to diverge in the resources they exploit[24,27]. For example, on Garden Island, Western Australian rock python (*Morelia spilota*) females weigh ten times as much as males, and the sexes differ in their dietary niches accordingly—males feed mostly on small prey such as lizards, birds or mice, whereas females primarily feed on large mammals such as possums and wallabies[28]. Similarly, size divergence could be achieved when sex-specific fitness increases at different phenotypic optima between males and females[23] —either because the sexes differ in their ecological niches due to intrinsic differences in energetic requirements (dimorphic niches) or because more than one optimal trait value exists for both sexes to potentially occupy (bimodal niches). Thus, natural selection can lead to the evolution of SSD without sexual conflict—as may be the case for pinnipeds[29], primates[30] and artiodactyls[31].

The actual evolutionary history of SSD can be complex and difficult to reconstruct. Sexual selection and ecological niche differentiation can act in synergy, and the evolution of SSD can predate, and even lead to, the evolution of mating systems, taking advantage of pre-existing differences in size[29]. Similarly, if, for example, selection acts to increase male size due to male–male combat, different selection pressures can simultaneously act to decrease female size for ecological reasons. Conversely, selection can lead to an increase or decrease in size in both sexes but differ in magnitude. Additionally, even if the size of only one sex is under selection the other sex may show a similar trend of size evolution, because of genetic correlations, but the magnitude of such change may be lower[32]. SSD can thus evolve as the change in size in one sex outpaces the other[33]. Therefore, despite the longstanding interest in elucidating the mechanisms underlying SSD across lineages, whether a dominant mechanism is generally involved across the majority of cases of SSD or even which of the sexes is subject to selection for SSD remain prevailing challenges in evolutionary biology.

Here, we use a phylogenetic comparative approach to study the drivers behind the evolution of SSD. We use the largest number of species that have been used in SSD studies to date (11,236 tetrapod species), encompassing large variation in SSD from extremely male-biased (for example, southern elephant seal, *Mirounga leonina*, where males are 5.3 times heavier than females) to extremely female-biased (for example, helmeted water toad, *Calyptocephalella gayi*, with females 21 times heavier than males). We estimate evolutionary change in male and female body sizes using a recently developed trait evolution model, the Fabric model[34], which incorporates variation in evolvability (the rate of trait evolution) and directional evolution (Methods). Briefly, the model can distinguish between directional changes (an increase or a decrease in the mean trait value in a descendant clade) and changes in evolvability (an increase or a decrease in the variance of the trait value in the descendant clade). We then test whether the observed distribution of SSD can be explained in the absence of directional evolution on male and female size, that is by variation in evolvability alone. Finally, we test whether the frequencies, magnitudes and directions of change in size of males and females are correlated with the magnitude and direction

(male versus female bias) of SSD. Taken together, we use this combination of phylogenetic comparative methods and macroevolutionary models to explore three broad questions about the evolution of SSD:

(1) How important is directional evolution in generating SSD? We expect that many of the observed SSD values result from directional evolution in the size of one sex causing intersexual divergence in body size, which is otherwise strongly correlated.

(2) How did the frequency of directional evolution vary among and between the sexes in different SSD classes (female- or male-biased) throughout the evolutionary history of tetrapods? Previous studies suggest that SSD may be, on average, biased towards males or females in large clades[35,36]. Using the Fabric model and binomial tests we assess whether dominant modes of SSD are the result of biases in the frequency of sex-specific directional shifts throughout the evolutionary history of tetrapods.

(3) Does sex-specific directional evolution tend to act more strongly in one of the sexes to drive the diversity of SSD? While SSD can be generated by selection acting on only one sex, it may also arise by selection operating on both sexes at different rates, magnitudes or directions (increasing or decreasing size)[33]. Previous research has suggested that selection may be stronger on males than on females[37,38]. However, this might be the opposite in female-biased clades, such as amphibians, where females are usually the larger sex[39]. We expect the magnitude and direction of evolution to be correlated with SSD: for example, stronger directional trends for larger male size or for smaller female size, would be correlated with more male-biased SSD, and vice versa for female-biased SSD.

## Results and discussion
### Variation in SSD among tetrapods
SSD is extremely common in tetrapods. Roughly two-thirds of species in our sample are sexually dimorphic (given an arbitrary definition of dimorphic sexes being >10% divergent in mass), yet these species are not randomly placed on the tetrapod tree of life. Some clades have strongly male- or female-biased SSD (Fig. 1 and Extended Data Fig. 1). Mammals are, on average, male-biased (mean SSD = −0.12; males ~13% heavier than females; 44.8% of species male-biased, 15.1% of species female-biased— similar to previous estimates[35]). Amphibians are, on average, strongly female-biased (mean SSD = 0.56; females ~75% heavier than males; 6.2% of species male-biased, 85.9% of species female-biased). Birds are, on average, monomorphic (mean SSD = −0.04; males ~4% larger; 29.2% of species male-biased, 11.8% of species female-biased). A recent study[36] did not use a cut-off for monomorphic species, but similarly found more than twice as many male-biased bird species as female-biased ones. Squamates (mean SSD = −0.01; males ~1% larger; 37.3% of species male-biased, 38.5% of species female-biased) are also monomorphic on average. However, both amphibians and squamates have higher variance in SSD (Fig. 1b; V = 0.25 and 0.15, respectively) than either mammals (V = 0.06) or birds (V = 0.02). Thus, birds are the only clade where most species are monomorphic (59% versus 40%, 24% and 8% of mammals, squamates and amphibians, respectively), while squamates are usually dimorphic, but similar proportions of species have female- and male-biased dimorphism. The most dimorphic species are usually either amphibians (93% of the top decile of female-biased species) or squamates (59% of the top decile of male-biased species; Fig. 1c).

### Observed patterns of SSD require directional evolution
To estimate how directional evolution on body size generates observed patterns of SSD, we used the directional random-walk Fabric[34] model. This recent macroevolutionary model allows the inference of directional evolution acting on a trait by estimating whether increases or decreases in mean trait value occurred along each branch in a phylogeny, while accommodating variability in the rate of trait evolution.

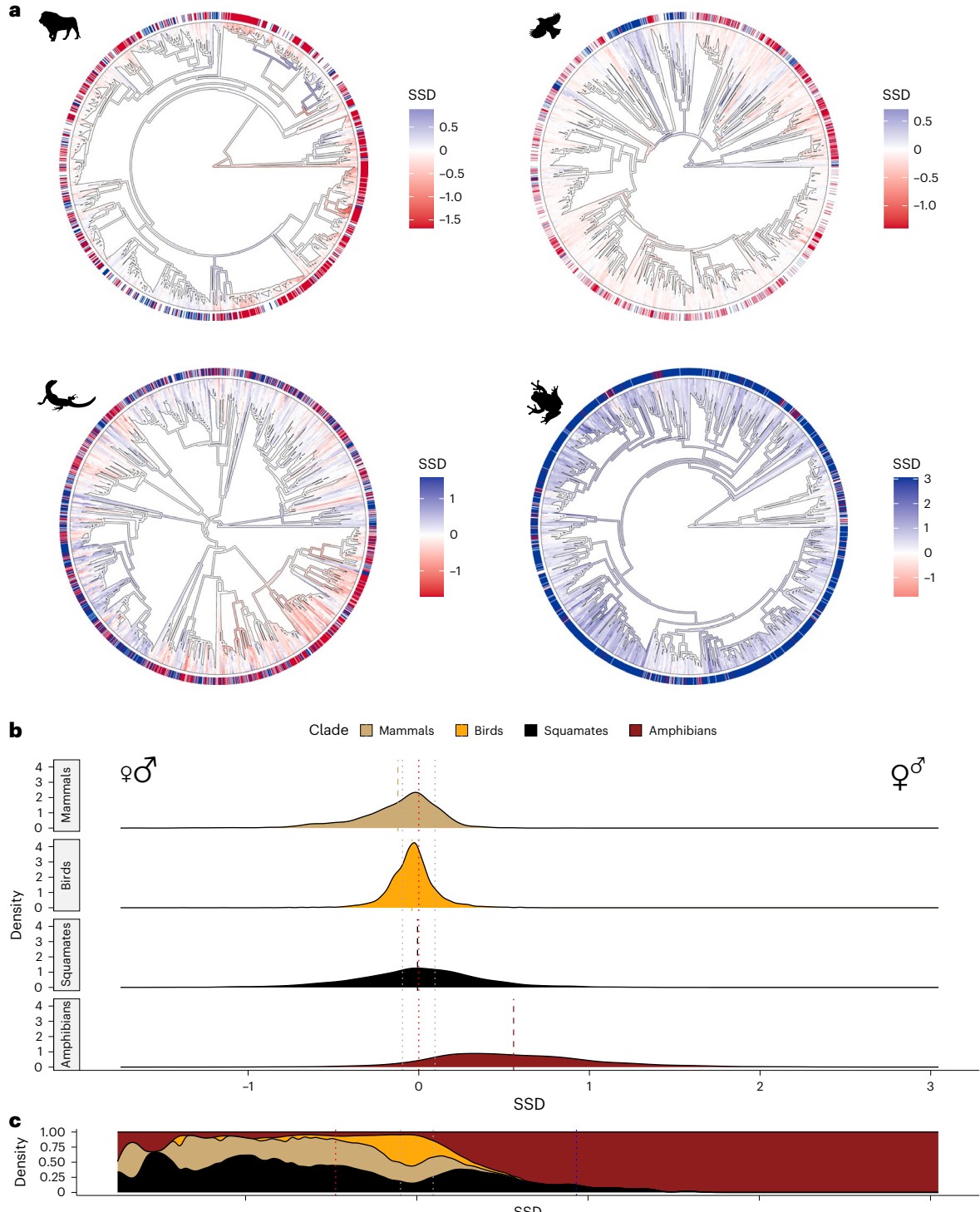

**Fig. 1 | Distribution of SSD across tetrapod clades. a**, SSD mapped onto the phylogenies of mammals, birds, squamates and amphibians. Surrounding the phylogenies, female-biased species in blue, male-biased species in red and monomorphic species (10% difference in size) in white. Along the branches, stronger colours denote higher values of SSD. **b**, Estimated kernel density plots of the distribution of SSD values in each of the four clades. Red dotted lines represent SSD = 0, grey dotted lines SSD values of 10% or less difference in size. Thick dashed lines represent the mean values for each clade. **c**, Density plot showing the relative proportion of species from the four clades for each value of SSD. Dotted grey lines represent 10% or less difference in size. Dotted blue and red lines represent the top deciles of female-biased and male-biased species, respectively. Icons from PhyloPic.org under a CC0 1.0 license: *Panthera leo*, Margot Michaud; *Turdus pillaris*, Sharon Wegner-Larsen; *Varanus komodoensis*, Steven Traver; *Hyla versicolor*, Will Booker.

We fitted Fabric models to estimate the number, magnitude (change in log units compared to the ancestral state) and direction (increasing or decreasing) of directional changes in body size for males and females in each clade, and the reconstructed ancestral states of male and female body size in each internal node in the phylogenies. Our models unequivocally rejected the Brownian motion, Ornstein–Uhlenbeck, accelerating–decelerating (delta) and evolvability (with and without trend) models for both male- and female-size evolution in all clades. The

**Table 1 | Parameter estimates for the best-fitting Fabric and global trend models**

| Clade | Sex | Number of directional shifts (% of total branches in tree) | Number of evolvability shifts | Global trend (95% HPD) | Brownian variance (95% HPD) | Root state estimate (95% HPD) | Inferred ancestral SSD |
|---|---|---|---|---|---|---|---|
| Mammals | Female | 299 (9.2%) | 149 | 1.009 (1.001–1.017) | 0.024 (0.013–0.044) | 168.5 (18.5–701.1) | −0.065 |
| | Male | 302 (9.3%) | 148 | 1.007 (0.999–1.011) | 0.019 (0.009–0.025) | 179.9 (17.7–729.6) | |
| Birds | Female | 571 (7.0%) | 324 | 1.005 (1.002–1.007) | 0.007 (0.005–0.008) | 302 (80.3–584.2) | 0.262 |
| | Male | 581 (7.1%) | 304 | 1.006 (1.001–1.009) | 0.008 (0.005–0.009) | 232.4 (48.7–510.4) | |
| Squamates | Female | 530 (8.5%) | 232 | 1.002 (1.001–1.003) | 0.007 (0.004–0.008) | 5.1 (1.6–11.7) | 0.101 |
| | Male | 547 (8.7%) | 250 | 1.003 (1.001–1.004) | 0.009 (0.006–0.010) | 4.6 (1.5–10.2) | |
| Amphibians | Female | 470 (9.9%) | 186 | 1.003 (0.997–1.005) | 0.019 (0.012–0.021) | 8 (1.5–31.7) | 0.175 |
| | Male | 474 (10.0%) | 227 | 1.003 (0.999–1.005) | 0.015 (0.011–0.019) | 6.7 (1.3–22.7) | |

The numbers of directional and evolvability shifts are taken as the numbers of shifts identified as significant in all four runs of BayesTraits. The median and highest posterior density (HPD) for the global trend parameter is represented as the inferred fold-change over the history of each clade from their respective common ancestor. Ancestral state estimates of body size in grams (median and HPD) for the root are shown, along with the inferred degree of SSD based on the ancestral state estimates.

strongest support was always for the Fabric model with a global trend (Supplementary Table 1), albeit with only marginally higher support for a global trend in some cases (for example, female size in amphibians). Directional shifts occurred more frequently than evolvability shifts in all clades. This means that directional changes in body size were the norm across all clades, implying net directional change in body size over macroevolutionary timescales throughout the evolutionary history of tetrapods. The number of directional shifts ranged from 299 (female mammal size) to 581 (male bird size) and global trends were consistently weak (Table 1). On the basis of estimated root states from the Fabric model we were also able to infer (with considerable uncertainty) that the common ancestor of mammals was approximately monomorphic, and that of birds, amphibians and squamates was slightly female-biased (Table 1).

The strong support for the Fabric model over evolvability-only models suggests that directional evolution is key to understanding the evolution and distribution of body size and SSD in tetrapods.

However, studies of species with extreme SSD suggest that body size divergence in the sexes can arise as a combination of strong selection on one sex, but not the other. For example, extremely large female-biased values of SSD in nephilid spiders may be driven by random evolution of female size coupled with selection towards a small optimal male size[40]. Therefore, the presence of directional shifts alone is not enough to understand the evolutionary history of SSD—we also need to know how many shifts occurred, and their locations and timing on the tree of life. We explore the relative importance of differential patterns of divergent evolution on sexes in the following two sections, first examining how the frequency of instances of directional changes in body size varies within and among sexes, and then assessing the magnitude of directional effects on the evolution of SSD.

**Directional shifts are more common in males**

We tested for sex biases in the frequency of directional evolution by summing the number of positive and negative changes for each sex in each SSD category, across all terminal and internal nodes from the fitted Fabric models[34] excluding directional changes that were not statistically supported (Methods). We found that, in all four tetrapod clades, females and males experience similar frequencies of directional changes in size (Fig. 2 and Extended Data Fig. 2)—we estimated 302 male versus 299 female changes in mammal size, 581 male versus 571 female changes in bird size, 547 male versus 530 female changes in squamate size and 474 male versus 470 female changes in amphibian size. None of these differences is statistically significant ($P = 0.51$–$0.52$ in all binomial tests), which is perhaps unsurprising given that body size in males and females is strongly positively correlated[32].

We next assessed whether, despite occurring in similar overall frequency, there were sex-specific patterns in directional changes.

SSD could arise if there is a directional shift in one sex and not the other or if there are directional shifts in opposite directions.

First, we examined whether there is a difference in the number of increases or decreases in size; that is, whether body size was more likely to become larger or smaller in one sex or the other. This allowed us to test whether, in each sex, there was an overall tendency for body size to decrease or increase. We found that there were significantly more increases than decreases in female size in monomorphic birds ($P = 0.015$). We found more increases than decreases in male size in male-biased mammals ($P < 0.001$), birds ($P = 0.021$), squamates ($P < 0.001$) and amphibians ($P < 0.001$). In female-biased taxa we found strong evidence for more decreases than increases in male size in all clades: mammals ($P = 0.003$), birds ($P = 0.036$), squamates ($P < 0.001$) and amphibians ($P = 0.012$).

Second, we examined whether directional changes of either type (increases or decreases) were more common in one sex or the other. This allowed us to test whether or not sexes diverged in their overall shifts—that is, whether one sex or the other was more likely to experience a particular type of directional shift. We found that there were significantly more male than female decreases in size in female-biased squamates ($P = 0.021$), more female than male increases in size in female-biased mammals ($P = 0.039$) and squamates ($P = 0.014$), more female than male decreases in size in male-biased squamates ($P < 0.001$), and more male than female increases in size in male-biased squamates ($P = 0.001$). All other comparisons were non-significant (Supplementary Tables 2–5).

**Sex-specific directional shifts are correlated with SSD**

From the Fabric models we extracted species and sex-specific measures of the magnitude of directional change, that is, downstream fold changes in mean mass of either sex. We then fitted phylogenetic generalized least squares (PGLS) models to estimate whether the magnitude of directional size changes estimated from the Fabric models was correlated with the magnitude of SSD in different species. The PGLS models show that SSD in mammals becomes more female-biased with greater decreases in male size, and SSD in amphibians becomes more male-biased with greater increases in male size (Fig. 3 and Table 2). Perhaps surprisingly, all other relationships were non-significant.

We then fitted phylogenetic multivariate response models (Fig. 4) to test the relative magnitude of body size changes of males and females. This enabled us to test whether directional changes are biased in favour of one sex or the other, depending on the overall direction of sexual dimorphism (male-biased, female-biased or monomorphic). We found that mammal females ($P_{ROPE} = 0$ where ROPE is region of practical equivalence) and squamate ($P_{ROPE} = 0$) females tend to increase more in size than do males in female-biased species, but not in birds ($P_{ROPE} = 0.566$) or amphibians ($P_{ROPE} = 0.871$). In all taxa

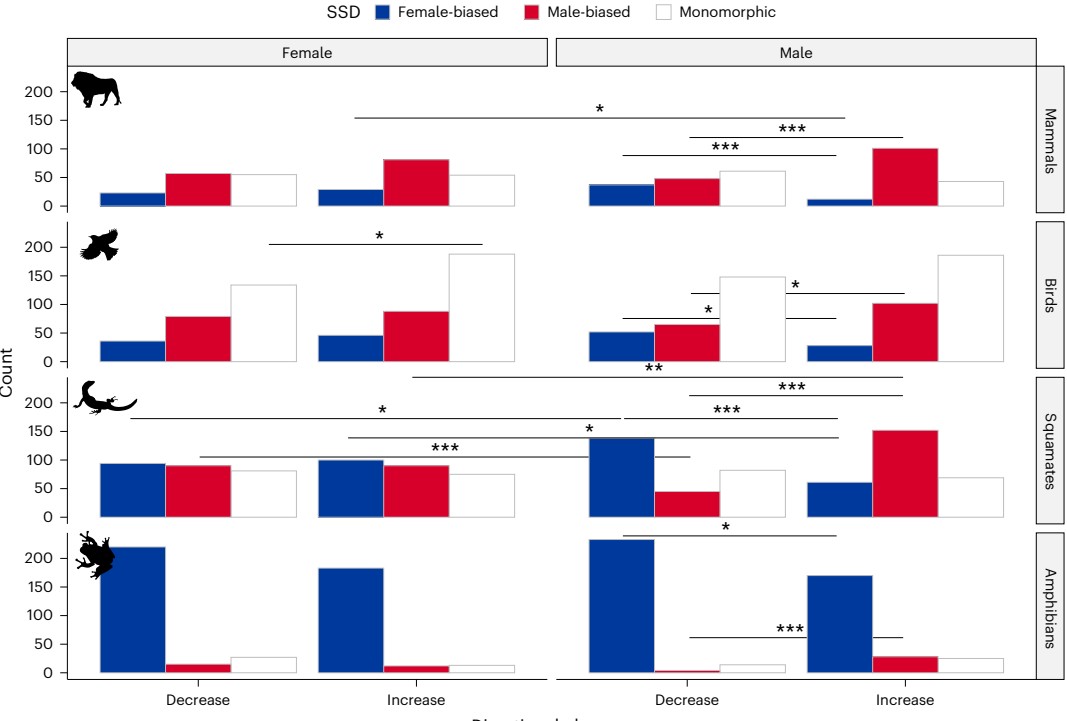

**Fig. 2 | Numbers of directional changes in each sex, in each of four tetrapod clades.** Directional changes are divided into decreasing or increasing, and counted in the three categories of SSD (blue for female-biased, red for male-biased, white for monomorphic). Statistically significant comparisons from two-sided binomial tests with a Benjamini–Hochberg FDR correction for *P* values are marked with horizontal lines and asterisks (*$P < 0.05$; **$P < 0.01$; ***$P < 0.001$). Icons from PhyloPic.org under a CC0 1.0 license: *Panthera leo*, Margot Michaud; *Turdus pillaris*, Sharon Wegner-Larsen; *Varanus komodoensis*, Steven Traver; *Hyla versicolor*, Will Booker.

(mammals, $P_{ROPE} = 0$; birds, $P_{ROPE} = 0.002$; squamates, $P_{ROPE} = 0$; amphibians, $P_{ROPE} = 0$) males increase more in size than females in male-biased species. In monomorphic amphibians only ($P_{ROPE} = 0$) male size increases more than female size ($P_{ROPE} = 0.267, 0.898, 0.772$ in mammals, birds and squamates, respectively). Thus, in almost all tetrapod clades, females tend to increase in size relative to males when species have female-biased SSD and males tend to increase in size relative to females in male-biased species (Fig. 4). In other words, female-biased species tend to be more female-biased than their ancestors and male-biased species tend to be more male-biased than their ancestors. However, this need not always be driven by an increase in size of the larger sex. For example, if selection on female body size decreases is accompanied by stronger selection on decreases in male size[37], the net result would still be female-biased SSD—and indeed our models infer this to be the case in some female-biased taxa, such as emballonurid bats (Supplementary Information 2).

**Inferring mechanisms of SSD evolution**
We find evidence that the evolution of SSD is more often driven by directional changes in male size, consistent with the idea that selection on body size is stronger in males[37,38]. Not only are directional changes in size more common in males (Fig. 2), but the magnitude of SSD is also mostly correlated with changes in male body sizes (Fig. 3 and Table 2). The exact relationships differ—in mammals, SSD becomes more positive (female-biased) in species where male size decreases, whereas in amphibians SSD becomes more negative (male-biased) in species where male size increases. Surprisingly, the pattern of the magnitude of SSD being driven more by changes in male size holds even for amphibians, despite females being larger in nearly all amphibian species (Fig. 1). These results seemingly contradict previous research that has suggested that the evolution of amphibian SSD is driven mostly by fecundity selection on females[8,39]. Strong selection pressures on

male body size in amphibians may arise from the physical constraints size enacts on mating calls[41]. However, we must stress that, while our results suggest that directional evolution resulting from female-specific selection is not a driver of SSD variation in amphibians, fecundity selection might still be a strong driver of the strongly female-biased ancestral SSD of amphibians. Thus, fecundity selection might maintain the class-wide pattern of female-biased SSD and strong male-specific selection (rather than relaxation of female-specific selection) might drive the few instances of male-biased SSD or monomorphism. Many anamniote taxa (including fishes and many invertebrate clades[42–45]) are also predominantly female-biased. We suspect that the evolution of amniote modes of reproduction (for example, their overall lower brood sizes[46]) may have relaxed fecundity selection and led amniotes to be either more variable or lean towards male-biased SSD compared to anamniotes (Fig. 1).

We suggest that the evolution of SSD is thus often driven by sex-specific selection, with lineages differing in which sex is under stronger selection, based on the interpretation of directional changes as being indicative of selection[34]. Often, but not always, sex-specific directional trends (indicating which sex is under stronger selection) are correlated with the prevalent direction of SSD in that clade (on average male- or female-biased).

Our analyses reveal that extant patterns in SSD were shaped by different sex-specific directional changes in size among and within tetrapod clades. These could be indicative of divergent agents of sex-specific selection throughout tetrapod evolutionary history. By estimating the amount of change in body size for females and males separately, we were able to identify that males typically experienced more frequent evolutionary changes in body size, which is indicative of stronger selection on males. However, selection on both female and male size has probably contributed to extant patterns of SSD in tetrapods.

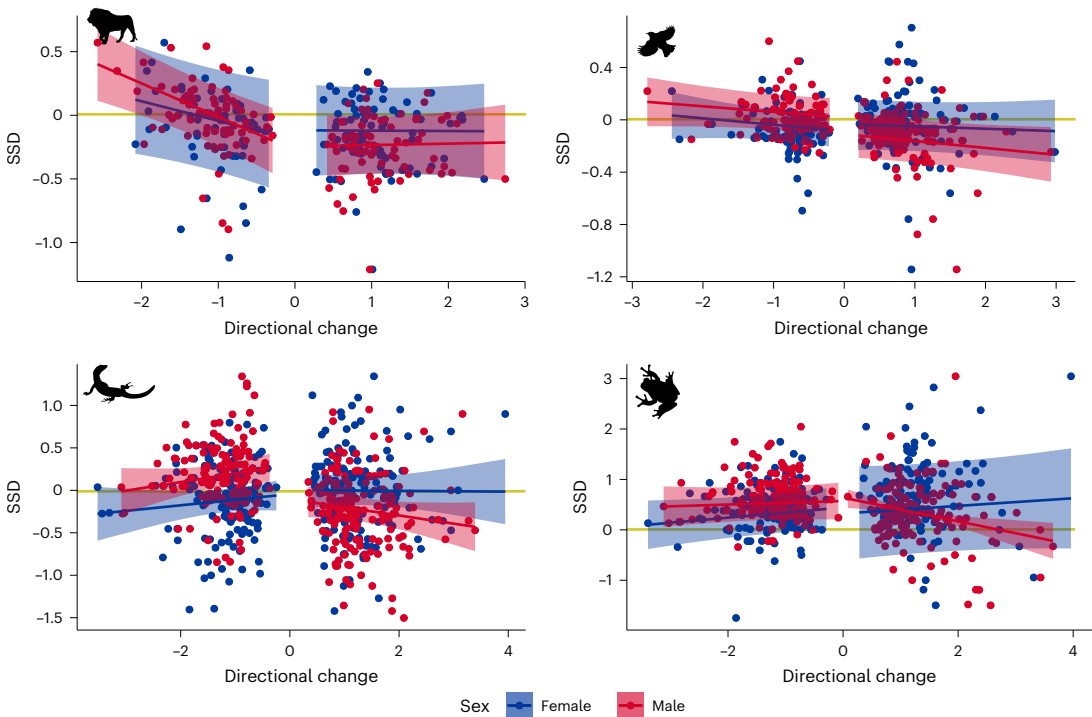

**Fig. 3 | Scatterplots showing the relationships from PGLS models of SSD regressed against directional change for each sex.** Females are shown in blue and males in red across all four tetrapod clades, with 95% confidence intervals represented by shading. Directional change describes the total amount of change along a branch that is attributable to directional effects on the natural log scale. Therefore, a directional change of 2 corresponds to a 7.38-fold increase in size. The yellow horizontal lines represent no SSD (females and males have equal sizes). Positive values (above the yellow line) represent species where females are larger than males and negative values (below the yellow line) represent species where males are larger than females. Icons from PhyloPic.org under a CC0 1.0 license: *Panthera leo*, Margot Michaud; *Turdus pillaris*, Sharon Wegner-Larsen; *Varanus komodoensis*, Steven Traver; *Hyla versicolor*, Will Booker.

**Table 2 | Summary of PGLS models of SSD regressed against absolute value of directional change in body size in each sex for four tetrapod clades**

| Clade | Sex | Directional change | n | Slope | λ | P |
|---|---|---|---|---|---|---|
| **Mammals** | Female | Decrease | 61 | −0.153 | 0.729 | 0.065 |
| | | Increase | 81 | −0.003 | 0.717 | 0.962 |
| | **Male** | **Decrease** | **77** | **−0.263** | **0.480** | **<0.001** |
| | | Increase | 75 | 0.012 | 0.407 | 0.851 |
| Birds | Female | Decrease | 97 | −0.492 | 0.545 | 0.251 |
| | | Increase | 143 | −0.018 | 0.863 | 0.611 |
| | Male | Decrease | 101 | −0.043 | 0.870 | 0.210 |
| | | Increase | 140 | −0.053 | 0.858 | 0.113 |
| Squamates | Female | Decrease | 152 | 0.066 | 0.181 | 0.292 |
| | | Increase | 150 | −0.005 | 0.227 | 0.943 |
| | Male | Decrease | 138 | 0.101 | 0.187 | 0.125 |
| | | Increase | 166 | −0.096 | 0.179 | 0.112 |
| **Amphibians** | Female | Decrease | 144 | 0.101 | 0.518 | 0.134 |
| | | Increase | 129 | 0.076 | 0.843 | 0.422 |
| | **Male** | Decrease | 139 | 0.035 | 0.439 | 0.582 |
| | | **Increase** | **147** | **−0.235** | **−0.014** | **0.003** |

The columns show the sample sizes, estimated slopes, λ values and the P values of the slope. Statistically significant slopes are in bold.

Our approach could further research on the evolutionary drivers of SSD. Rather than using proxies to infer specific mechanisms, we directly test the evolutionary trajectories of body size through fitting complex trait evolution models and comparison to null models to infer the existence and direction of selection. Thus, we can narrow the field of potential mechanisms to test for. As an example, pinnipeds (seals, walruses and sea lions) display some of the most extreme male-biased SSD in our dataset. Pinnipeds also have polygynous mating systems, with intense male–male combat for territories. Much research has focused on examining the evolution of SSD in pinnipeds under the selective pressure of male–male combat, but evidence to support this idea has often been lacking[29,47]. Using our approach we find that variation in pinniped SSD does not appear to be driven by increasing male size, as would be expected if male–male combat drove the evolution of SSD—rather, both male and female sizes increased early in the evolution of pinnipeds and even earlier, before their lineage split from bears (Ursidae; also with strongly male-biased SSD; Supplementary Information 2). The only evidence for male-specific changes to have occurred in pinnipeds is for increase in male size in the ancestor of the (extremely male-biased) elephant seals (*Mirounga*) and in the male-biased grey seal (*Halichoerus grypus*). These are among the most dimorphic species in their family—and in mammals in general. We also detected evidence for reduction in male size in the ancestor of the genus *Pusa*, among the smallest and least dimorphic species of pinnipeds. Male-biased SSD therefore appears to be ancestral in pinnipeds. Thus, we lend support to the suggestion that male-biased SSD in pinnipeds evolved before the evolution of their characteristic mating systems[29], and instead may be driven by other processes, and perhaps even a sequential combination of multiple forms of selection[48].

## Conclusions

In summary, we have shown that SSD shows great variation in tetrapods not only in its direction and magnitude, but also in its drivers. Greater and more frequent evolutionary changes in male sizes appear to be the norm among tetrapods—yet the exact mechanisms in which SSD

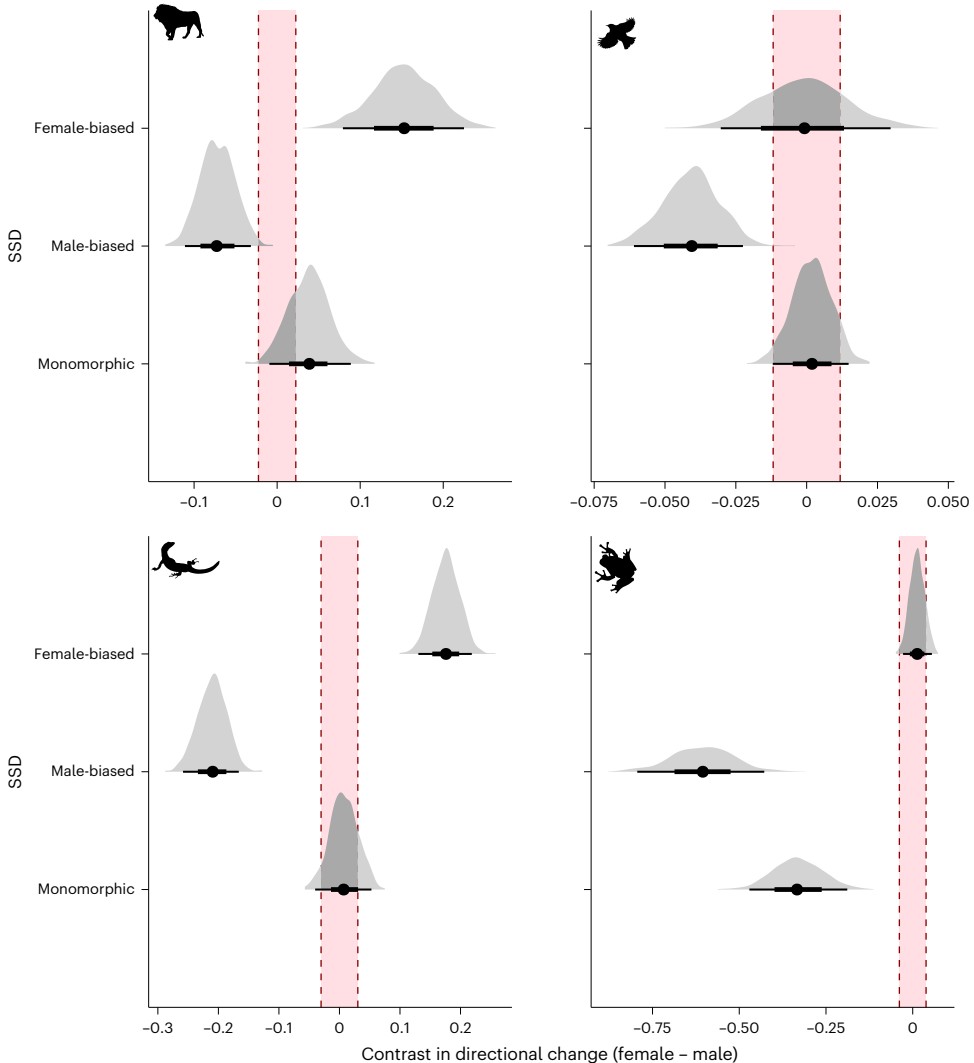

**Fig. 4 | Posterior distributions of contrasts in directional changes (female change − male change) for species in the three SSD categories for four tetrapod clades.** Positive values indicate females increase in size compared to males and negative values indicate the opposite (although both sexes might experience increase or decrease in size compared to the ancestral value). The red shaded area represents the ROPE, where contrasts cannot be distinguished from zero. Darker grey regions of each posterior distribution represent areas that overlap the ROPE. Icons from PhyloPic.org under a CC0 1.0 license: *Panthera leo*, Margot Michaud; *Turdus pillaris*, Sharon Wegner-Larsen; *Varanus komodoensis*, Steven Traver; *Hyla versicolor*, Will Booker.

evolves can differ between clades. The lack of universal mechanisms generating SSD is perhaps unsurprising and may prove a complex challenge for future research into the evolution of SSD. We posit that our approach offers a way forward by directly inferring directional changes in body size in each sex, offering insight on when and where selection on body size occurred. Thus, we may be more informed when devising and testing our hypotheses on the selection pressures underlying the evolution of SSD on a case-by-case basis.

## Methods

### Data collection

We compiled a global-scale database on mean male and female body sizes for four tetrapod clades (amphibians, squamate reptiles, birds and mammals) from published sources and museum specimens[11,49–51]. We supplemented these with newly collected data for snakes and some lizard, bird and mammal species. SSD data were usually calculated from sex-specific mean size of adults and occasionally from the midpoint of the adult size range (Supplementary Information 1). To make the datasets comparable, we used mass (g) as our proxy for body size[52,53]. We transformed newly collected snout–vent length (SVL)

data for squamates from SVL to mass using clade-specific allometric equations[52]. We used recently published time-calibrated phylogenies (using trees that only included species with genetic data and not species placed via taxonomy-based polytomy resolvers) for each of the four major tetrapod clades we examined[54–57] and pruned them to only include sampled species. We used the published consensus trees for amphibians[54], mammals[55] and squamates[56]. For birds[57], we downloaded the full 10,000 tree posterior generated using the Hackett backbone for species with genetic data (6,670 species) and constructed a maximum clade credibility consensus tree using TreeAnnotator v.1.8.4 (ref. 58). Our final dataset comprised 2,369 species of amphibians, 4,098 species of birds, 1,633 species of mammals and 3,136 species of squamates: total 11,236 tetrapod species, that is, roughly a third of all tetrapod species.

For each species, we calculated SSD using the following equation:

$$\text{SSD} = \ln(mass_f / mass_m) \tag{1}$$

such that positive values of SSD represent female-biased species (females larger than males) and negative values of SSD represent

male-biased species (males larger than females). This measure of SSD has the benefits of being symmetric[59] and, being on a log scale, allowing easy visualization of large variation in SSD. We used an arbitrary cut-off of 10% difference in mass (absolute SSD value ~0.09) to categorize species as either monomorphic (at or below the cut-off) or dimorphic (above it).

### Inferring sex-specific evolution

For each of the four tetrapod clades, we used BayesTraits v.4.1.1 (ref. 60) to fit the Fabric model of continuous trait evolution with a global trend[34] for log-transformed male and female size. The Fabric model estimates two different types of changes in trait evolution dynamics along the phylogeny—directional changes ($\beta$), which either increase or decrease a trait value along a branch and affect all downstream descendants, and evolvability changes ($v$), which either increase or decrease the variance of a trait in a clade, but do not affect the mean trait values. Thus, directional changes can be inferred to represent directional selection on a trait, whereas evolvability changes represent differences in the rate of trait evolution (different potential to explore trait space). The Fabric model thus enables us to quantify the phylogenetic positions and magnitude of directional changes in the evolution of female and male body sizes—and therefore to infer if observed SSD is driven by selection acting on one sex (for example, female size increase coupled with no change in male size or vice versa), both (for example, female size decrease coupled with male size increase) or neither (both sexes show no directional changes and observed SSD derives from random walk).

The numbers, values and positions of both types of changes, as well as parameters of Brownian motion variance, global trends in trait evolution and ancestral trait estimates, were identified using an MCMC sampling algorithm. For each clade we ran four MCMC chains for $10.5 \times 10^7$ generations, sampling every $10^5$ generations and discarding the first $5 \times 10^6$ generations as burn-in, under seven different models: Brownian motion, Ornstein–Uhlenbeck, accelerating or decelerating evolution (delta), evolvability (equivalent to Fabric without directional trends), evolvability with a global trend, Fabric without a global trend and Fabric with a global trend. We then estimated marginal likelihood for each model by using a stepping-stone sampler implemented in BayesTraits using 1,000 stones. We selected the best model for each sex of each clade with the highest mean marginal likelihood across all four runs. In all four clades this was Fabric with global trend (Supplementary Table 1).

We followed recommendations in ref. 34 by setting a Weibull prior ($\kappa = 1.5$ and $\lambda = 1.1$) on $\beta \times t$ effects (directional changes along a branch of length $t$), which was found to be an adequate prior for body mass and a Gamma prior ($\alpha = 1.2$ and $\beta = 5$) on evolvability changes[34]. We set Gamma priors on the ancestral trait estimates to reflect reasonable estimates for each clade: $\alpha = 25.6$ and $\beta = 0.25$ for mammals and birds (centred on a mean estimate of ~601 g and ranging between ~7 g and 60 kg) and $\alpha = 4.8$ and $\beta = 0.5$ for squamates and amphibians (centred on a mean estimate of ~11 g and ranging between ~0.1 g and 1.1 kg). We then ensured proper mixing of the chains and ran all downstream analyses in R v.4.1.0 (ref. 61). We visually assessed trace plots from the four runs using the mcmc_trace function in the bayesplot package v.1.8.1 (ref. 62) to ensure convergence, combined all runs using the combine.mcmc function from the runjags package v.2.2.1.7 (ref. 63) and calculated effective sample sizes for the combined runs using the effectiveSize function from the coda package v.0.19.4 (ref. 64).

To determine where directional branches occurred along the phylogenies, we selected the directional changes which exceeded the 2 s.d. criterion described in ref. 34 in all four runs for each sex. We identified whether each change was positive (body mass increased along the branch) or negative (body mass decreased along the branch). We then ran several analyses to test if SSD evolution is likely to be driven by sex-specific selection by addressing the frequency of shifts ((1) and (2) below) and the magnitude of shifts ((3) and (4)):

(1) To examine if the number of positive and negative directional changes differed within sexes in each category of SSD (female- or male-biased)—for example, are there more positive than negative changes in male body size in male-biased species—we performed binomial tests. For these tests we summed the number of positive and negative changes for each sex, in each category of SSD, across all nodes in the phylogenies ($n = 601, 1,152, 1,077$ and 944 for mammals, birds, squamates and amphibians, respectively), while discarding changes that did not exceed the 2 s.d. criterion described above ($n = 2,664, 7,043, 5,194$ and 3,793 for mammals, birds, squamates, and amphibians, respectively). $P$ values were adjusted for multiple comparisons using a Benjamini–Hochberg false discovery rate (FDR) correction.

(2) To examine if the number of directional positive and negative changes differed between sexes in each category of SSD (for example, are there more positive changes in body size in male than female size in male-biased species?), we ran binomial tests. For these tests we summed the number of positive and negative changes for each sex in each category of SSD across terminal branches only as above.

(3) To assess whether SSD increases proportionally with directional change (for example, does SSD become more male-biased as males increase more in size?) we fitted PGLS regressions of SSD against the value of directional change (in log units, that is, fold changes in mass) in each sex and each type of change (negative or positive), using the gls function from the nlme package v.3.1.152 (ref. 65), while estimating the maximum likelihood value of $\lambda$. For these tests we used only the terminal branches in each phylogeny which exceeded the 2 s.d. criterion described above and treated the change along the terminal branch leading to each species as the response value. In each analysis, we pruned the tree to include only these species ($n = 139, 237, 289$ and 268 for female mammals, birds, squamates and amphibians, respectively, and $n = 179, 302, 392$ and 376 for male mammals, birds, squamates and amphibians, respectively).

(4) To test whether one sex experiences larger directional change than the other in different SSD categories (for example, do males, on average, increase in size more than females in male-biased species?), we fitted phylogenetic multivariate response models using the MCMCglmm package v.2.32 (ref. 66). For these analyses, we used all branches in the phylogenies and treated the degree of directional change along the branch leading to each species as the continuous response value, after omitting changes that did not exceed the 2 s.d. criterion described above. We used male and female directional changes as multivariate Gaussian response values, set all priors to their default values (nu = 0, $V = 1$, alpha.mu = 0, alpha.$V$ = 0), ran chains for 1 million generations sampling every 1,000 generations and discarded the first 10% as burn-in. We ascertained that acceptance ratios were >0.25, visually assessed trace plots and calculated effective sample sizes to ensure proper mixing and exploration of parameter space. We then estimated the significance of contrasts between female and male directional change by calculating the proportion of the posterior distribution of contrasts that does not lie within the ROPE (region of practical equivalence) using the p_rope function from the bayestestR package v.0.11.5 (ref. 67).

### Reporting summary

Further information on research design is available in the Nature Portfolio Reporting Summary linked to this article.

## Data availability

Data used to run the analyses are available via Figshare at https://doi.org/10.6084/m9.figshare.20416245 (ref. 68). Results from Bayes-Traits runs are available via Figshare at https://doi.org/10.6084/m9.figshare.20416083 (ref. 69).

## Code availability

Code to run the analyses are available via Figshare at https://doi.org/10.6084/m9.figshare.20416245 (ref. 68).

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

## Acknowledgements

We are extremely grateful to A. Meade for assisting us with estimating ancestral states when running the Fabric model in BayesTraits. A.S. was supported by a Royal Society grant RGF\EA\181082 to G.H.T. G.H.T. was supported by a Royal Society University Research Fellowship (URF\R\180006).

## Author contributions

A.S. devised the study, performed the analyses and led the writing. N.C., S.M., G.M., D.P.-D. and G.H.T. contributed data, advised on methodology and study design and contributed to revision of the manuscript.

## Competing interests

The authors declare no competing interests.

## Additional information

**Extended data** is available for this paper at https://doi.org/10.1038/s41559-024-02600-8.

**Correspondence and requests for materials** should be addressed to Alex Slavenko or Gavin H. Thomas.

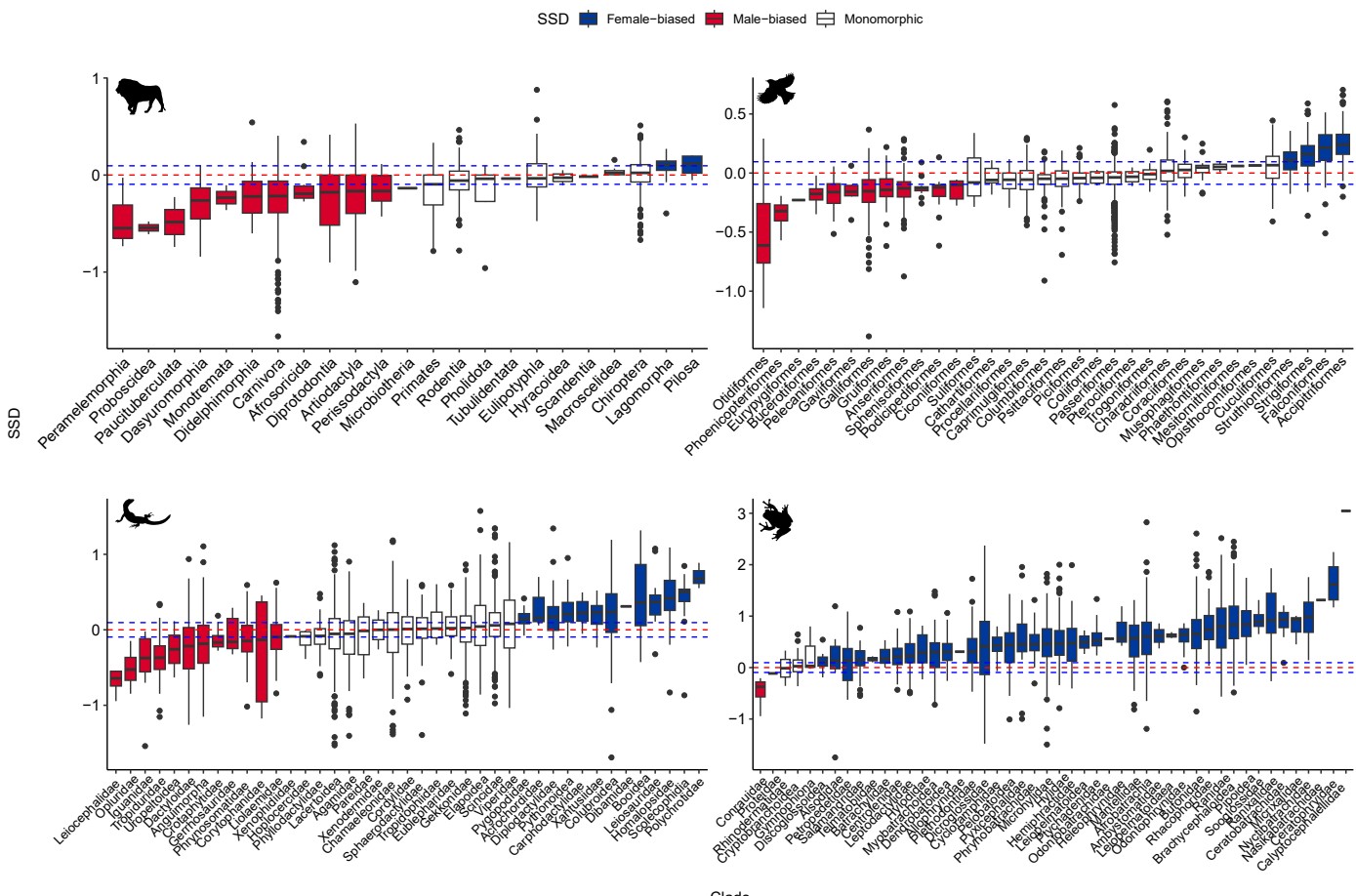

**Extended Data Fig. 1 | Distribution of SSD within tetrapod clades.** Boxplots showing the distributions of SSD in each order of mammals (n = 1,633 species and 23 orders) and birds (n = 4,098 species and 34 orders) and family of squamates (n = 3,136 species and 41 families) and amphibians (n = 2,369 species and 48 families). The red dashed line represents SSD of 0, and the blue dotted lines represent the cut-off between monomorphic and dimorphic species, at 10% difference in size (absolute SSD value - 0.09). Boxplot are coloured according to the median SSD value in each clade – red for male-biased, blue for female-biased, and white for monomorphic. Boxplots are centred on the median value of each group, with the top and bottom bounds of the box representing the 75th and 25th percentiles, respectively, and the top and bottom whiskers representing the maximum and minimum values within 1.5 times the interquartile range, respectively. Icons from PhyloPic.org under a CC0 1.0 license: *Panthera leo*, Margot Michaud; *Turdus pillaris*, Sharon Wegner-Larsen; *Varanus komodoensis*, Steven Traver; *Hyla versicolor*, Will Booker.

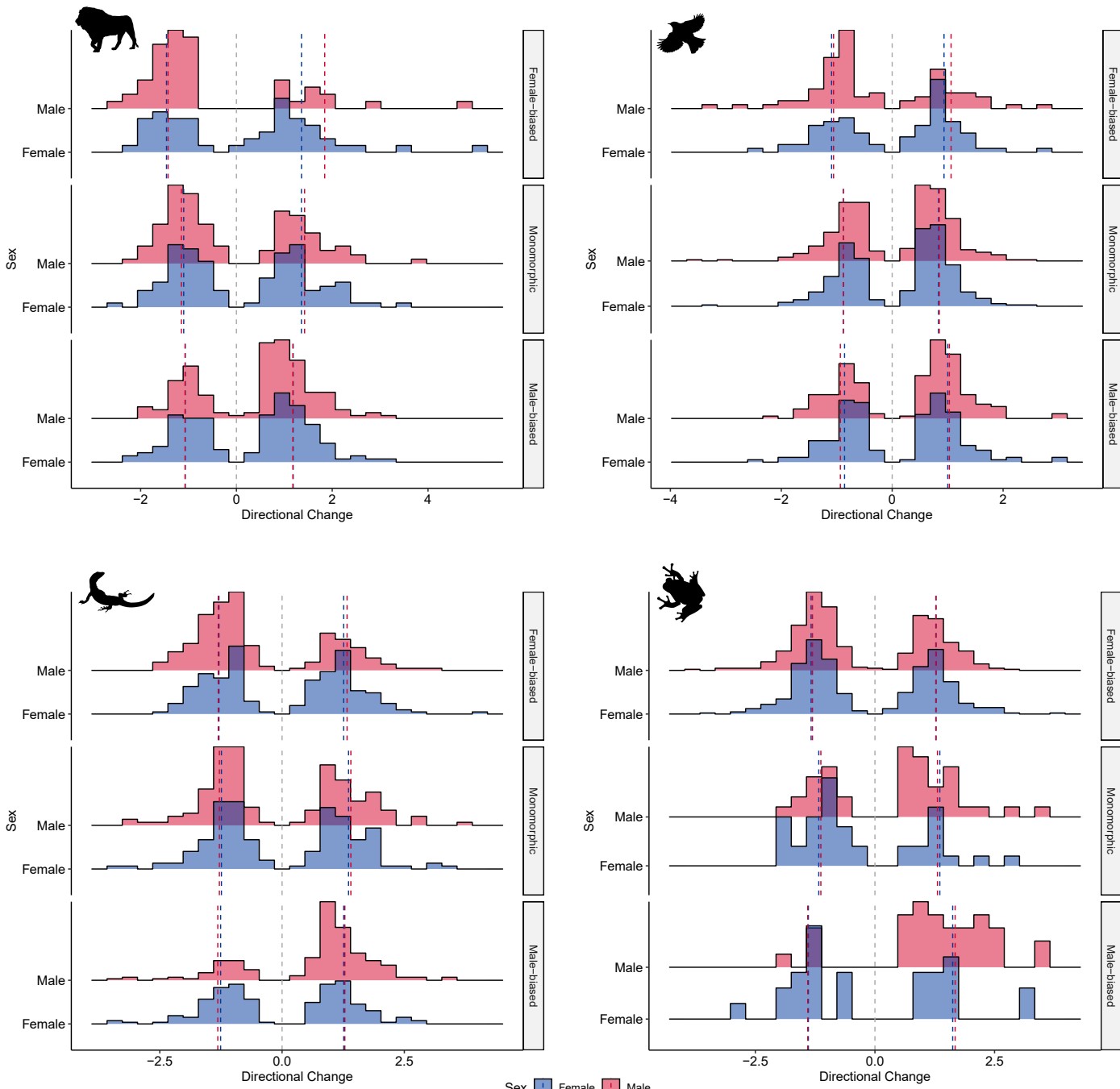

**Extended Data Fig. 2 | Numbers of directional changes in body size throughout the evolutionary history of tetrapods.** Histograms showing the distributions of directional changes (either increasing or decreasing) for each sex in each of four tetrapod classes, in three different SSD categories. Dashed grey lines represent 0 directional change, and blue and red dashed lines represent mean directional change for females and males, respectively. Icons from PhyloPic.org under a CC0 1.0 license: *Panthera leo*, Margot Michaud; *Turdus pillaris*, Sharon Wegner-Larsen; *Varanus komodoensis*, Steven Traver; *Hyla versicolor*, Will Booker.

# Reporting Summary

## Statistics

For all statistical analyses, confirm that the following items are present in the figure legend, table legend, main text, or Methods section.

| n/a | Confirmed | |
|---|---|---|
| ☐ | ☒ | The exact sample size (*n*) for each experimental group/condition, given as a discrete number and unit of measurement |
| ☒ | ☐ | A statement on whether measurements were taken from distinct samples or whether the same sample was measured repeatedly |
| ☐ | ☒ | The statistical test(s) used AND whether they are one- or two-sided<br>*Only common tests should be described solely by name; describe more complex techniques in the Methods section.* |
| ☐ | ☒ | A description of all covariates tested |
| ☐ | ☒ | A description of any assumptions or corrections, such as tests of normality and adjustment for multiple comparisons |
| ☐ | ☒ | A full description of the statistical parameters including central tendency (e.g. means) or other basic estimates (e.g. regression coefficient) AND variation (e.g. standard deviation) or associated estimates of uncertainty (e.g. confidence intervals) |
| ☒ | ☐ | For null hypothesis testing, the test statistic (e.g. *F*, *t*, *r*) with confidence intervals, effect sizes, degrees of freedom and *P* value noted<br>*Give P values as exact values whenever suitable.* |
| ☐ | ☒ | For Bayesian analysis, information on the choice of priors and Markov chain Monte Carlo settings |
| ☒ | ☐ | For hierarchical and complex designs, identification of the appropriate level for tests and full reporting of outcomes |
| ☐ | ☒ | Estimates of effect sizes (e.g. Cohen's *d*, Pearson's *r*), indicating how they were calculated |

*Our web collection on statistics for biologists contains articles on many of the points above.*

## Software and code

Policy information about availability of computer code

Data collection | No software was used to collect data included in this study.

Data analysis | All data were analysed in BayesTraits v4.1.1 and R v4.1.0. Code to run the analyses can be found on the following Figshare repository: 10.6084/m9.figshare.20416245.

For manuscripts utilizing custom algorithms or software that are central to the research but not yet described in published literature, software must be made available to editors and reviewers. We strongly encourage code deposition in a community repository (e.g. GitHub). See the Nature Portfolio guidelines for submitting code & software for further information.

## Data

Policy information about availability of data

All manuscripts must include a data availability statement. This statement should provide the following information, where applicable:
- Accession codes, unique identifiers, or web links for publicly available datasets
- A description of any restrictions on data availability
- For clinical datasets or third party data, please ensure that the statement adheres to our policy

Data and code used in this manuscript are freely available on Figshare. Code and data used to run the analyses can be found on the following repository: 10.6084/m9.figshare.20416245. Results from BayesTraits runs can be found in the following repository: 10.6084/m9.figshare.20416083.

# Research involving human participants, their data, or biological material

Policy information about studies with [human participants or human data](). See also policy information about [sex, gender (identity/presentation), and sexual orientation]() and [race, ethnicity and racism]().

| | |
|---|---|
| Reporting on sex and gender | *Use the terms sex (biological attribute) and gender (shaped by social and cultural circumstances) carefully in order to avoid confusing both terms. Indicate if findings apply to only one sex or gender; describe whether sex and gender were considered in study design; whether sex and/or gender was determined based on self-reporting or assigned and methods used. Provide in the source data disaggregated sex and gender data, where this information has been collected, and if consent has been obtained for sharing of individual-level data; provide overall numbers in this Reporting Summary. Please state if this information has not been collected. Report sex- and gender-based analyses where performed, justify reasons for lack of sex- and gender-based analysis.* |
| Reporting on race, ethnicity, or other socially relevant groupings | *Please specify the socially constructed or socially relevant categorization variable(s) used in your manuscript and explain why they were used. Please note that such variables should not be used as proxies for other socially constructed/relevant variables (for example, race or ethnicity should not be used as a proxy for socioeconomic status). Provide clear definitions of the relevant terms used, how they were provided (by the participants/respondents, the researchers, or third parties), and the method(s) used to classify people into the different categories (e.g. self-report, census or administrative data, social media data, etc.) Please provide details about how you controlled for confounding variables in your analyses.* |
| Population characteristics | *Describe the covariate-relevant population characteristics of the human research participants (e.g. age, genotypic information, past and current diagnosis and treatment categories). If you filled out the behavioural & social sciences study design questions and have nothing to add here, write "See above."* |
| Recruitment | *Describe how participants were recruited. Outline any potential self-selection bias or other biases that may be present and how these are likely to impact results.* |
| Ethics oversight | *Identify the organization(s) that approved the study protocol.* |

Note that full information on the approval of the study protocol must also be provided in the manuscript.

# Field-specific reporting

Please select the one below that is the best fit for your research. If you are not sure, read the appropriate sections before making your selection.

☐ Life sciences   ☐ Behavioural & social sciences   ☒ Ecological, evolutionary & environmental sciences

For a reference copy of the document with all sections, see [nature.com/documents/nr-reporting-summary-flat.pdf]()

# Ecological, evolutionary & environmental sciences study design

All studies must disclose on these points even when the disclosure is negative.

| | |
|---|---|
| Study description | Macroevolutionary analysis of the evolution of SSD in tetrapods, using phyogenetic comparative methods, species-specific mass estimates for males and females, and published phylogenies. |
| Research sample | Male and female mass data for 2,369 species of amphibians, 4,099 species of birds, 1,633 species of mammals, and 3,136 species of squamates; total 11,237 tetrapod species. |
| Sampling strategy | Data were widely collected from published datasets and the literature, to encompass as large as possible a dataset on global tetrapod body sizes. |
| Data collection | Data were collected from published datasets and the literature. |
| Timing and spatial scale | Global scale; temporal scale irrelevant. |
| Data exclusions | Data were collected for all available species which were also represented in published phylogenies. |
| Reproducibility | Irrelevant - no experiments conducted. |
| Randomization | Irrelevant - no experiments conducted. |
| Blinding | Irrelevant - data were collected from published datasets and literature. |

Did the study involve field work?   ☐ Yes   ☒ No

# Reporting for specific materials, systems and methods

We require information from authors about some types of materials, experimental systems and methods used in many studies. Here, indicate whether each material, system or method listed is relevant to your study. If you are not sure if a list item applies to your research, read the appropriate section before selecting a response.

## Materials & experimental systems

| n/a | Involved in the study |
|-----|------------------------|
| ☒ ☐ | Antibodies |
| ☒ ☐ | Eukaryotic cell lines |
| ☒ ☐ | Palaeontology and archaeology |
| ☒ ☐ | Animals and other organisms |
| ☒ ☐ | Clinical data |
| ☒ ☐ | Dual use research of concern |
| ☒ ☐ | Plants |

## Methods

| n/a | Involved in the study |
|-----|------------------------|
| ☒ ☐ | ChIP-seq |
| ☒ ☐ | Flow cytometry |
| ☒ ☐ | MRI-based neuroimaging |

## Plants

| | |
|---|---|
| Seed stocks | Irrelevant - study was on animals. |
| Novel plant genotypes | Irrelevant - study was on animals. |
| Authentication | Irrelevant - study was on animals. |

