## [Peer Review File · Nature Ecology & Evolution]

Evolution of sexual size dimorphism in tetrapods is driven by varying patterns of sex-specific selection on size

Corresponding Author: Dr Alex Slavenko

Version 0:

Decision Letter:

27th September 2022

Dear Alex,

Your manuscript entitled "Evolution of sexual size dimorphism in tetrapods is driven by varying patterns of sex-biased selection on size" has now been seen by two reviewers, whose comments are attached. The reviewers have raised a number of concerns which will need to be addressed before we can offer publication in Nature Ecology & Evolution. We will therefore need to see your responses to the criticisms raised and to some editorial concerns, along with a revised manuscript, before we can reach a final decision regarding publication.

We therefore invite you to revise your manuscript taking into account all reviewer and editor comments. Please highlight all changes in the manuscript text file in Microsoft Word format.

* If you have not done so already please begin to revise your manuscript so that it conforms to our Article format instructions at <http://www.nature.com/natecolevol/info/final-submission>. Refer also to any guidelines provided in this letter.

Link Redacted

Nature Ecology & Evolution is committed to improving transparency in authorship. As part of our efforts in this direction, we are now requesting that all authors identified as 'corresponding author' on published papers create and link their Open Researcher and Contributor Identifier (ORCID) with their account on the Manuscript Tracking System (MTS), prior to acceptance. ORCID helps the scientific community achieve unambiguous attribution of all scholarly contributions. You can create and link your ORCID from the home page of the MTS by clicking on 'Modify my Springer Nature account'. For more

information please visit www.springernature.com/orcid.

[redacted]

Reviewer expertise:

Reviewer #1: evolution of SSD

Reviewer #2: comparative phylogenetics

Reviewers' comments:

Reviewer #1 (Remarks to the Author):

I am not familiar with FABRIC model. Therefore, I am only able to offer a general comment. I found the approach very useful and the conclusions challenging. The manuscript consolidates some old ideas but at the same time proposes original viewpoints that will open new perspectives in the research of SSD evolution.

My only specific suggestion is that the authors should review the mechanisms proposed by Slatkin (1984) to explain the evolution of SSD under natural selection. It is not only inter-sexual competition but also two other processes that deserve being discussed in this manuscript, considering that the conclusion is that natural selection appears to have an important role in the evolution of tetrapod SSD, especially in mammals. Perhaps you may find useful the following articles I wrote in relation to this topic.

1. Cassini MH (2020) A mixed model of the evolution of polygyny and sexual size dimorphism in mammals. *Mammal Review* 50: 112-120
2. Cassini MH (2020) Sexual size dimorphism and sexual selection in primates *Mammal Review*, 50: 231-239
3. Cassini, M. H. (2022). Evolution of sexual size dimorphism and sexual segregation in artiodactyls: the chicken or the egg?. *Mammalian Biology*
4. Cassini MH (2017) Role of fecundity selection on the evolution of size sexual dimorphism in mammals. *Animal Behaviour* 128:1-4.

Marcelo H. Cassini

Reviewer #2 (Remarks to the Author):

Overall, I was interested to read this manuscript. I think it's a topic that will be of broad interest, and uses an extensive dataset alongside modern, up-to-date methods. However, I unfortunately cannot recommend the manuscript for publication in its current form. The reason for this is two-fold; I will explain in more detail below.

Firstly, there is a lack of clarity regarding the analyses which makes it very difficult to understand what exactly the authors have done. Importantly, it is not clear how exactly the tests used in the statistical analyses were carried out. The authors describe how they identify individual branches (lines 299-304) but then go on to say that they run PGLS models of SSD against the absolute value of directional change in each sex and each type of change. What is the data here? Is it branch-wise or species-level? If it is species level is this calculated only on the terminal branch leading to the species or is it a path-wise measure? If it only includes branches which fit the two standard deviation criterion, then we'd expect some huge reductions in sample sizes (implied in the extended data tables) which does not seem to be the case in Figure 2.

The binomial tests examining number of positive and negative directional changes also suffer from the same lack of clarity – number of changes across a whole tree? A tree limited to only the male-biased taxa? Branches that fit the criterion? Terminal branches? This needs to be outlined much more clearly and it must be described how this data is classified. Furthermore, it is difficult to know what these tests are studying – is this not just studying whether there is sexual size dimorphism? Please see below with regards to the rationale behind analyses.

The PGLS analyses of SSD against directional change are stated to study change in log units of body mass (e.g., Figure 2) but this cannot be correct. For example, are we expected to believe that there are repeatedly body mass changes exceeding 10g observed in amphibians? I expect these axes are standardized, but this is just another example of the lack of clarity.

The simulation procedure does not seem to add much in its current state – it might be interesting to see what the simulated data gives in terms of the fabric model but as far as I understand this was not done?

Importantly, I think it needs to be outlined as to how exactly applying the fabric model provides a novel way to approach this

in relation to what has been done before. It currently feels a little bit like the authors had a set of analyses that they have quickly re-worked to use the recently published fabric model without a comprehensive understanding of why. However, what are the expectations with regards to this model? What about the post-hoc analyses? How would the authors expect their results to look? This will also help readers interpret what could potentially be rather complicated results and figures (see above).

In its current state, it is impossible to understand exactly what the authors did and why, and it is therefore difficult to assess the quality of the manuscript.

Secondly, it is also not entirely obvious what the overall goal of the manuscript is. It currently suffers from a vagueness that feels exploratory. I think it would be in the authors' best interests to go through the manuscript and clearly outline the questions they are trying to address and how what they are doing goes towards achieving that. For example, I'd like to see some expectations set out in the introduction – while there is a good background to the topic, it is not clear what the authors are seeking to understand.

Finally, and a rather minor point with respect to those outlined above, the first paragraph of the results section is simply a list of descriptive statistics about the data. I don't think this is necessary at all and in fact may benefit being moved to the figure captions or to relevant parts of the discussion. Otherwise, as this paragraph currently lacks any biological interpretation, it is very dense on data description and should be removed.

*****END*****

Version 1:

Decision Letter:

9th November 2022

Dear Alex,

Your revised manuscript entitled "Evolution of sexual size dimorphism in tetrapods is driven by varying patterns of sex-biased selection on size" has now been seen by the second reviewer, whose comments are attached. The reviewer believes the paper improved in clarity but has raised a number of concerns which will need to be addressed before we can offer publication in Nature Ecology & Evolution. We will therefore need to see your responses to the criticisms raised and to some editorial concerns, along with a revised manuscript, before we can reach a final decision regarding publication.

Although all points are important and must be fully addressed, we want to emphasize the need to analyse all the branches rather than the terminal ones only and the need to estimate the significance of the global beta in the fabric model, and indicated by the reviewer. I should stress that this is your last chance to satisfy the reviewer.

We therefore invite you to revise your manuscript taking into account all reviewer and editor comments. Please highlight all changes in the manuscript text file in Microsoft Word format.

* If you have not done so already please begin to revise your manuscript so that it conforms to our Article format instructions at <http://www.nature.com/natecolevol/info/final-submission>. Refer also to any guidelines provided in this letter.

Link Redacted

Nature Ecology & Evolution is committed to improving transparency in authorship. As part of our efforts in this direction, we are now requesting that all authors identified as 'corresponding author' on published papers create and link their Open Researcher and Contributor Identifier (ORCID) with their account on the Manuscript Tracking System (MTS), prior to acceptance. ORCID helps the scientific community achieve unambiguous attribution of all scholarly contributions. You can create and link your ORCID from the home page of the MTS by clicking on 'Modify my Springer Nature account'. For more information please visit www.springernature.com/orcid.

[redacted]

Reviewers' comments:

Reviewer #2 (Remarks to the Author):

I believe this manuscript has improved its overall clarity regarding the PGLS analyses and binomial tests. However, there are several issues outstanding that have not been addressed satisfactorily that preclude my ability to recommend this manuscript for publication in its current form. I will outline these here.

While the two sets of binomial tests are different – I note the now clear explanations, but the difference between questions two and three could be a little unclear to the unfamiliar reader. For example, what is the difference between these two expectations: “stronger selection on male size will be correlated with male-biased SSD” and “selection may be stronger on males than on females... expect male shifts in size to be greater in male size-biased clades”? This needs to be laid out much more clearly for the reader to understand.

“A simple variable rate random walk null model” (L132-134) is quite contrary – this is not at all a simple model! I appreciate the authors explanation in their response to reviewers, but I would make clear that these simulations are accounting for observed heterogeneity in the rate of body size evolution and SSD. I would recommend some of the wording in the author’s response be moved to the text. The results and discussion of the manuscript needs to be restructured to focus on the statistical analysis – the authors themselves state that these simulations are intended to be supplementary to the fabric model analyses yet are the first focus of the results in the current manuscript.

The arguments laid out with regards to SSD and selection in lines 142-156 are flawed and incomplete. Firstly, using the simulated distributions as evidence that extreme values need not be selective is misleading. Indeed, the first example given (nephilid spiders), explicitly invokes selection on (male) body size! There is also good evidence in the literature that high rates of change can be associated with selective pressures (even in the absence of long-term directionality). The arguments laid out by the authors rely heavily on the OU model which is not representative of the model the authors use (Pagel et al 2022). The purpose seems to be to lead into the simulations whose entire purpose is to look at the change in variance across SSD which is not relevant to the questions of interest. Overall, without a significant overhaul of both the approach and justification, the simulation procedure in its current form is not adding much and should be excluded – at no detriment to the manuscript.

I’m sorry, but the authors’ explanation regarding Figure 2 is not satisfactory, and it still does not make sense. The x axis cannot be measured in percent change. It seems very surprising to me to say that in most branches of the phylogenetic tree that body mass is more than doubling (or halving) – these are very large changes in mass to be occurring so frequently. I believe this is a mis-understanding of Pagel et al (2022) which describes the beta values in the context of “fold” changes – “a slope of 0.0056 is equivalent to a 1.013 ($10^{0.0056}$) fold increase in un-logged size”. For a beta value of 1 on even a natural log scale, it would be equivalent to a 2.71 fold increase in un-logged size – this is very large. This needs more clarification.

The authors choice to study terminal branches rather than all branches is justified by the fact that they do not infer ancestral SSD. However, ancestral states can readily be derived from the output of the fabric model. Regardless, studying terminal branches is an unusual approach to studying evolutionary change – it is ignoring the millions of years of shared evolutionary change among species and only looking at the last few million years or so of independent evolution. In fact, looking at only terminal branches could be very misleading as the authors themselves acknowledge (L114-115) that SSD is not randomly distributed across the phylogenetic tree. Because of this, it is expected that changes along internal branches leading to clades are likely to have profound impacts on the evolutionary distribution of SSD across tetrapods.

It comes to my attention here that the authors do not determine the significance of the global beta in the fabric model – while they discuss the global trend models it is absolutely essential to compare the statistical fit of a model with and without global trends – for example, using marginal Lhs. If a global trend is found to be non-significant, all subsequent analyses are rendered meaningless – the betas will be very different between the two models.

Minor points:

- The null hypothesis (line 342) should specify clearly that SSD would be derived from a non-directional or unbiased random walk – the fabric model with betas is also a random walk model.
- On line 328, the authors state that the ancestral trait estimates are identified – but they are only estimating the root estimate here, not the trait estimates for all ancestors. This should be clarified (although see above).
- Did the authors test the effect of priors on their analysis? While they suggest they followed recommendations in Pagel et al (L331-335), the priors recommended in this paper were specifically for log10 mammalian body size measured in kg. Here, the authors are studying natural logged body size of tetrapods measured in grams.
- On the subject of priors, the authors should also specify the reasoning behind the gamma prior on the ancestral trait estimate – what does this distribution centre on in units of body size – and why? Is it based on fossil estimates for the size of the basal tetrapod?

*****END*****

Version 2:

Decision Letter:

1st October 2024

Dear Alex,

Your manuscript entitled "Evolution of sexual size dimorphism in tetrapods is driven by varying patterns of sex-specific selection on size" has now been seen by a new reviewer. Unfortunately, Reviewer #2 was not able to re-review and we found a replacement reviewer to look at your responses to their comments. This reviewer has done that but has also raised new issues as you can see in the comments below my signature. We will therefore need to see your responses to the criticisms raised along with a revised manuscript, before we can reach a final decision regarding publication.

I should stress that points 1 and 3 from reviewer #2 need further work. For point 3, you will need to remove the simulations from the manuscript.

* If you have not done so already please begin to revise your manuscript so that it conforms to our Article format instructions at <http://www.nature.com/natecolevol/info/final-submission>. Refer also to any guidelines provided in this letter.

Link Redacted

Nature Ecology & Evolution is committed to improving transparency in authorship. As part of our efforts in this direction, we are now requesting that all authors identified as 'corresponding author' on published papers create and link their Open Researcher and Contributor Identifier (ORCID) with their account on the Manuscript Tracking System (MTS), prior to acceptance. ORCID helps the scientific community achieve unambiguous attribution of all scholarly contributions. You can create and link your ORCID from the home page of the MTS by clicking on 'Modify my Springer Nature account'. For more information please visit www.springernature.com/orcid.

[redacted]

Reviewers' comments:

Reviewer #3 (Remarks to the Author):

Summary

In this paper, the authors attempt to identify some of the causal mechanisms behind the evolution of SSD. They use PCMs and a dataset of tetrapods to estimate directional changes in body size evolution, which are then used to infer differences in size changes between sexes. They suggest several findings, including that directional selection on body size is typical for tetrapods, the patterns of SSD necessitate an explanation of directional evolution, and that selection on size generating SSD is typically stronger in males. Overall, I believe the authors have tackled an interesting problem using a rich dataset. Nonetheless, I have concerns about the methodology employed by the authors and whether some of the claims listed above are substantiated by their results.

Major Comments

My primary concern with this work is the authors' choice to analyze male and female body size separately. I understand this decision allows the exploration of whether shifts in SSD are associated with shifts in male or female body size, but I believe this choice may lead to undesirable consequences. First, male and female body sizes are not independent; they are highly correlated with one another, and SSD is highly correlated with body size in general (Fairbairn 1997). However, in all downstream statistical tests in this manuscript, they seem to be treated as independent variables, such that shifts in male body size are examined and compared independently to shifts in female body size. Given the highly correlated nature of these variables, I would like the authors to explain why this would not result in inflated significance. If the body size of the lineage increases in general, we can expect both male and female body sizes to increase. Modeling shifts independently ignores this fact. One solution could be to model SSD directly. Treating SSD as a single variable (rather than as two independent variables for male and female body sizes) should resolve this problem. Once shifts in SSD are identified, the authors could then examine shifts in male and female body sizes separately to qualitatively assess when shifts are driven by males or females.

The authors conduct several binary tests. The repeated tests will lead to inflated type I error, and some form of correction (e.g., Bonferroni) should be applied. Additionally, the section beginning at Line 202 should report effect sizes in addition to p-values.

One of the conclusions the authors make is that SSD in tetrapods "cannot be explained without directional evolution." This conclusion is based on their modeling results using Fabric, which compares directional selection against a model of shifting variance and a more standard Brownian motion model. I do not believe this is a comprehensive enough model set to make that claim. Two potential alternatives are a model of stabilizing selection and a model where trait disparity has decreased over time. These are the Ornstein-Uhlenbeck and early burst models from Hansen (1997) and Harmon et al. (2010). These are standard phylogenetic comparative models that have been used to discuss body size evolution. Without including these more standard models, particularly an OU-type process, it is difficult to trust that directional selection is necessary for SSD. I acknowledge that it may be challenging to directly compare the Fabric model to an OU process, but an OU model should result in a variance pattern similar to the directional model. Perhaps a simulation assessment, as the authors did later in the paper, would be appropriate.

I believe the authors are too liberal in their use of the term "selection." The models they are fitting, like all PCMs, are phenomenological. They describe processes not based on first principles but on potential explanations of patterns. For example, while Brownian motion has a parameter, one would not interpret sigma as the rate of genetic drift. This is because drift, like directional selection, acts on fundamentally different time scales. Thus, while the authors repeatedly make claims about sex-specific selection, I do not believe this is accurate. This language should be considered more carefully. The authors use more appropriate terminology elsewhere in the paper when they discuss directional shifts rather than directional selection.

Minor Comments

The claims made from lines 186 to 201 suggest that directional selection tends to result in species that are monomorphic. I was missing some information about the parameter values and what they represent. A directional trend is interesting, but what exactly is being measured? What is the magnitude of the effect in numerical terms?

In the section "Selection tends to act more frequently on males," I think it would be helpful to explain what is meant by "within and between sexes." In the methods, you include "e.g., are there more positive than negative changes in male body size in male-biased species?" This sort of explanation could be added when presenting the results.

Figure 1B: It may be worth adding a graphic that indicates whether values on the x-axis are male- or female-biased. For example, use traditional male and female symbols where, on the rightmost side of the graph (where females > males), the symbol for females is larger than that for males, and vice versa on the leftmost side of the figure. The meaning of the values is also not mentioned in the figure caption.

Line 98: The brief model description could be worded more clearly. When I first read the sentence, I wasn't sure whether the Fabric model distinguishes between directional changes and evolvability, or if it distinguishes between increases and decreases in both directional changes and evolvability.

Line 232: Does "decreasing male size" mean that as male size decreased, X happened, or does it mean that as the magnitude of negative male size shifts increased, X happened? Based on Figure 3, I would infer the latter, but the text suggests the former. My phrasing "magnitude of negative male size shifts increased" is unclear, so perhaps the authors could clarify this.

Line 279: Some of the material in the conclusion would be better placed in the results and discussion. Some new information is presented (e.g., the paragraph starting at line 285), and I think this could form its own section in the discussion, keeping the conclusion more concise.

Line 408: What is meant by "log units"? Is this the log of the parameter value? What are the units of the parameter value?

Line 421: Be clear that the response variable of directional change is continuous. The phrasing initially led me to think it was categorical.

Comments on responses to Reviewer #2:

Point 1: The choice to examine number and direction of shifts was based on point 1 of reviewer 2. In response, the authors changed their approach to look at direction and number of shifts. I do not believe this choice addressed the lack of clarity indicated by the reviewer. Their choice of verbiage comparing within and between sexes as well as within and between males/ females was still confusing even if their new approach made it easier to distinguish between hypotheses.

Point 2: the authors additional analysis and changes to the text satisfactorily addressed the concerns of the reviewer.

Point 3: I agree with the reviewer that the simulations are not necessary for this manuscript. I also do not believe the parameter estimates from the model are sufficiently discussed in the text despite the new table. I also agree with the reviewers comments about the OU model being more appropriate to examine "selection" (though I strongly feel this terminology is not appropriate). The simulations do not provide sufficient support to overcome this critique in my opinion.

Point 4: This was addressed sufficiently.

Point 5: The authors have addressed reviewer 2s concerns.

Point 6: This was addressed by the authors.

Minor points: Sufficiently addressed by the authors.

*****END*****

Version 3:

Decision Letter:

23rd October 2024

Dear Alex,

Thank you for submitting your revised manuscript "Evolution of sexual size dimorphism in tetrapods is driven by varying patterns of sex-specific selection on size" (NATECOLEVOL-220817210C). It has now been seen again by Reviewer 3 and their comments are below. The reviewers find that the paper has improved in revision, and therefore we'll be happy in principle to publish it in Nature Ecology & Evolution, pending minor revisions to satisfy the reviewers' final requests and to comply with our editorial and formatting guidelines.

[redacted]

Reviewer #3 (Remarks to the Author):

The authors have addressed all of my concerns adequately and I will respect their disagreement on point 1.

Reviewer #3 (Remarks on code availability):

Since the code is mainly in one long script, it could use some additional comments to make clear what each section does.

Reviewers' comments:

Reviewer #1 (Remarks to the Author):

I am not familiar with FABRIC model. Therefore, I am only able to offer a general comment. I found the approach very useful and the conclusions challenging. The manuscript consolidates some old ideas but at the same time proposes original viewpoints that will open new perspectives in the research of SSD evolution.

Thank you for the kind words!

My only specific suggestion is that the authors should review the mechanisms proposed by Slatkin (1984) to explain the evolution of SSD under natural selection. It is not only intersexual competition but also two other processes that deserve being discussed in this manuscript, considering that the conclusion is that natural selection appears to have an important role in the evolution of tetrapod SSD, especially in mammals. Perhaps you may find useful the following articles I wrote in relation to this topic.

1. Cassini MH (2020) A mixed model of the evolution of polygyny and sexual size dimorphism in mammals. *Mammal Review* 50: 112-120
2. Cassini MH (2020) Sexual size dimorphism and sexual selection in primates *Mammal Review*, 50: 231-239
3. Cassini, M. H. (2022). Evolution of sexual size dimorphism and sexual segregation in artiodactyls: the chicken or the egg?. *Mammalian Biology*
4. Cassini MH (2017) Role of fecundity selection on the evolution of size sexual dimorphism in mammals. *Animal Behaviour* 128:1-4.

Marcelo H. Cassini

We thank Dr. Cassini for rightfully pointing out that Slatkin invoked two additional mechanisms – we have now added a discussion of them in the Introduction (lines 67-73) and have incorporated the suggested references in the manuscript to bolster the general discussion of mechanisms generating SSD.

Reviewer #2 (Remarks to the Author):

Overall, I was interested to read this manuscript. I think it's a topic that will be of broad interest, and uses an extensive dataset alongside modern, up-to-date methods. However, I unfortunately cannot recommend the manuscript for publication in its current form. The reason for this is two-fold; I will explain in more detail below.

We thank the reviewer for their extensive comments. We have revised the manuscript to add clarity to the methods and the main text, and hope they will find this new version satisfactory.

Firstly, there is a lack of clarity regarding the analyses which makes it very difficult to understand what exactly the authors have done. Importantly, it is not clear how exactly the tests used in the statistical analyses were carried out. The authors describe how they identify individual branches (lines 299-304) but then go on to say that they run PGLS models of SSD against the absolute value of directional change in each sex and each type of change. What is the data here? Is it branch-wise or species-level? If it is species level is this calculated only on the terminal branch leading to the species or is it a path-wise measure? If it only includes branches which fit the two standard deviation criterion, then we'd expect some huge

reductions in sample sizes (implied in the extended data tables) which does not seem to be the case in Figure 2.

We have added clarification that the PGLS analyses were species-level, considering only species for which the directional change along the terminal branch leading to the species exceeded the two standard deviationw criterion (lines 381-384). The reviewer is correct that this resulted in large reductions in sample size, which we also now report (lines 384-387). However, please note that Figure 2 still contains the reduced sample sizes, since the analyses were done on hundreds of species (rather than thousands in the original, unpruned dataset).

The binomial tests examining number of positive and negative directional changes also suffer from the same lack of clarity – number of changes across a whole tree? A tree limited to only the male-biased taxa? Branches that fit the criterion? Terminal branches? This needs to be outlined much more clearly and it must be described how this data is classified. Furthermore, it is difficult to know what these tests are studying – is this not just studying whether there is sexual size dimorphism? Please see below with regards to the rationale behind analyses.

We thank the reviewer for this important request for more clarification. We have now added more detail to the description of our methodology in the text. To answer this particular question – the binomial tests examine the number of changes across terminal branches in the tree, since we refrain from making inference on ancestral SSD, and so cannot compare between internal branches which do not have data on whether these taxa are female- or male-biased (or monomorphic). We compare, in each class, the numbers of directional changes of each type (positive or negative) between both sexes and all categories of SSD. In other words, in the first analysis (within sexes), the test compares number of increases and decreases, and this is done for each category of SSD and each sex separately. In the second analysis (between sexes), the test compares number of changes in males and females, and this is done for each category of SSD and each type of change (increase or decrease) separately. Since changes are estimated for each sex separately, they represent the change in body size compared to the ancestral value – and therefore contrasting the numbers of changes does not study where sexual size dimorphism occurs, but rather where evolutionary changes in body size of either sex occur. The purpose of these tests is to see how these evolutionary changes in body size correlate with the observed patterns of SSD. We have now clarified the methodology and the rationale in the text (lines 97-112; 337-346; 365-370).

The PGLS analyses of SSD against directional change are stated to study change in log units of body mass (e.g., Figure 2) but this cannot be correct. For example, are we expected to believe that there are repeatedly body mass changes exceeding 10g observed in amphibians? I expect these axes are standardized, but this is just another example of the lack of clarity.

The axes are not standardised in this plot. Since the input data for the FABRIC models was in logged body size, directional changes $\beta \times t$ represent changes in the same units (logged body size) along branches. Thus, the directional changes we report are in log units, and represent fold-changes – e.g., a directional change of 1 represents a doubling in body mass compared to the ancestral mass. We have re-written this legend and omitted “of body mass [g]” in the hopes it will make it clearer that these are fold changes.

The simulation procedure does not seem to add much in its current state – it might be

interesting to see what the simulated data gives in terms of the fabric model but as far as I understand this was not done?

The simulation procedure was intended to be supplementary to the Fabric model analyses, by simulating evolvability changes alone (as empirically estimated by the Fabric model) without the directional change effects. This provided us with a way to generate a null expectation of what the evolution of SSD would look like in the examined taxa without directional changes. In theory it's also possible to fit Fabric models to the simulated phylogenies and compare fit and estimated parameters – however, this is not feasible due to the time-consuming nature of the analyses (we estimate fitting Fabric models to 8000 datasets [1000 simulations * 2 sexes * 4 classes] would require months, if not years).

Importantly, I think it needs to be outlined as to how exactly applying the fabric model provides a novel way to approach this in relation to what has been done before. It currently feels a little bit like the authors had a set of analyses that they have quickly re-worked to use the recently published fabric model without a comprehensive understanding of why. However, what are the expectations with regards to this model? What about the post-hoc analyses? How would the authors expect their results to look? This will also help readers interpret what could potentially be rather complicated results and figures (see above).

We thank the reviewer for this useful comment. We realise that the main text was lacking important detail about the Fabric model and why we decided to use, which was otherwise provided in part in the extended Methods section. We have now added clarification and justification for the use of the Fabric model (lines 178-182; 337-346).

In its current state, it is impossible to understand exactly what the authors did and why, and it is therefore difficult to assess the quality of the manuscript.

We hope the clarifications we have added and edits we have made to the text will make our goals and reasoning clearer.

Secondly, it is also not entirely obvious what the overall goal of the manuscript is. It currently suffers from a vagueness that feels exploratory. I think it would be in the authors' best interests to go through the manuscript and clearly outline the questions they are trying to address and how what they are doing goes towards achieving that. For example, I'd like to see some expectations set out in the introduction – while there is a good background to the topic, it is not clear what the authors are seeking to understand.

We thank the reviewer for this recommendation. We have added a paragraph to the end of the Introduction to clearly explain our research questions and expectations.

Finally, and a rather minor point with respect to those outlined above, the first paragraph of the results section is simply a list of descriptive statistics about the data. I don't think this is necessary at all and in fact may benefit being moved to the figure captions or to relevant parts of the discussion. Otherwise, as this paragraph currently lacks any biological interpretation, it is very dense on data description and should be removed.

We understand the reviewer's concerns about this paragraph, but we respectfully disagree about its importance. We think these descriptive statistics about the data are important both to give the reader an understanding of the structure of the data we analyse, and more importantly to quantify how SSD differs within and among tetrapods. We therefore think placing this information at the start of the results is important to provide the necessary biological and statistical context to the reader.

Reviewer #2 (Remarks to the Author):

I believe this manuscript has improved its overall clarity regarding the PGLS analyses and binomial tests. However, there are several issues outstanding that have not been addressed satisfactorily that preclude my ability to recommend this manuscript for publication in its current form. I will outline these here.

While the two sets of binomial tests are different – I note the now clear explanations, but the difference between questions two and three could be a little unclear to the unfamiliar reader. For example, what is the difference between these two expectations: “stronger selection on male size will be correlated with male-biased SSD” and “selection may be stronger on males than on females... expect male shifts in size to be greater in male size-biased clades”? This needs to be laid out much more clearly for the reader to understand.

We acknowledge the lack of clarity here and that, as written, the two questions could not be easily distinguished. We also note that it was not straightforward to map the questions directly to the analyses. We have attempted to remove ambiguity here by reformulating questions two (lines 110-116) and three (lines 117-126) around the frequency of directional changes compared to the magnitude of change. Question two now addresses the *frequency* of directional changes, and their sex-specific biases, throughout tetrapod evolutionary history (i.e. including all branches in the respective phylogenies). This question then maps to the two sets of binomial tests. Question three now focuses on the *magnitude* and *direction* of change in male and female body size, and the extent to which directional change can predict the mode of SSD (male-biased, female-biased, monomorphic). This question maps to the PGLS and MCMCgImm analyses.

“A simple variable rate random walk null model” (L132-134) is quite contrary – this is not at all a simple model! I appreciate the authors explanation in their response to reviewers, but I would make clear that these simulations are accounting for observed heterogeneity in the rate of body size evolution and SSD. I would recommend some of the wording in the author’s response be moved to the text. The results and discussion of the manuscript needs to be restructured to focus on the statistical analysis – the authors themselves state that these simulations are intended to be supplementary to the fabric model analyses yet are the first focus of the results in the current manuscript.

Thanks for raising this point. We have added further clarification of the complexity of the models to the results section by removing the term “simple” (we agree, they are not simple models, and we just meant simpler than the directional model) and clarifying that the models accommodate heterogeneity but not directional evolution (lines 174-178). We also directed the reader to the relevant section of the methods (Modelling SSD) where the simulation procedure is explained in detail.

Regarding the results and discussion, we agree that the Fabric model results should have been more prominent. We have added new text (lines 151-170) that describes the core outcomes of the Fabric analyses prior to discussing the simulation outputs. Together, these results set the scene for the subsequent analyses that test how the frequency and

magnitude of sex specific selection on size can explain variation in SSD among tetrapod species.

The arguments laid out with regards to SSD and selection in lines 142-156 are flawed and incomplete. Firstly, using the simulated distributions as evidence that extreme values need not be selective is misleading. Indeed, the first example given (nephilid spiders), explicitly invokes selection on (male) body size! There is also good evidence in the literature that high rates of change can be associated with selective pressures (even in the absence of long-term directionality). The arguments laid out by the authors rely heavily on the OU model which is not representative of the model the authors use (Pagel et al 2022). The purpose seems to be to lead into the simulations whose entire purpose is to look at the change in variance across SSD which is not relevant to the questions of interest. Overall, without a significant overhaul of both the approach and justification, the simulation procedure in its current form is not adding much and should be excluded – at no detriment to the manuscript.

We concur that we have not explained ourselves well enough but respectfully disagree regarding relevance. We have overhauled this section and place much stronger emphasis on the results of the Fabric model, including a new table (Table 1) to highlight the parameter estimates from the Fabric model. The strong support for directional evolution prompts us to investigate the effects of inclusion of directional evolution on phenotypic outcomes. The point of the simulated null models then is to assess whether the observed variation in SSD could arise by chance without the need to infer any generalised directional trends. We feel that being able to reject a null model is highly relevant and adds to our understanding of how directional selection influences the evolution of SSD.

We agree that our discussion here conflated directional selection with evolution around adaptive optima (as in the OU model). We have therefore re-focused this section more explicitly on the outcomes and implications of the Fabric model and our simulations. We now highlight that directional evolution limits diversity at the clade level. We retain the nephilid spider example to illustrate how selection on one sex, but not the other, can potentially lead to sexual size dimorphism. We use this example to link this section more strongly to the next two sections where we examine whether directional evolution tends to act more strongly on one sex than the other.

I'm sorry, but the authors' explanation regarding Figure 2 is not satisfactory, and it still does not make sense. The x axis cannot be measured in percent change. It seems very surprising to me to say that in most branches of the phylogenetic tree that body mass is more than doubling (or halving) – these are very large changes in mass to be occurring so frequently. I believe this is a mis-understanding of Pagel et al (2022) which describes the beta values in the context of “fold” changes – “a slope of 0.0056 is equivalent to a 1.013 ($10^{0.0056}$) fold increase in un-logged size”. For a beta value of 1 on even a natural log scale, it would be equivalent to a 2.71 fold increase in un-logged size – this is very large. This needs more clarification.

We apologise that this remained unclear. The values are in natural log units and describe the change in size occurring along a branch that is attributable to directional effects. We have re-written the figure legend to make this clear. As the reviewer suggests, a value of 1 would

suggest a 2.71 fold increase in unlogged size along a single branch. However, while this may seem large, Pagel et al's (2022) figure 2 highlights examples of >100 fold reductions in size and ~95-fold decreases in size in their analyses of mammals. The range of fold-increases in size that we observe is therefore consistent with previous work. We also note that the figure only includes data for significant directional change. We identified 299-581 significant shifts: 7-10% of the total branches in the tree. Thus, body mass is not more than doubling (or halving) in most branches of the phylogenetic tree – but in a relatively small minority. The example of a fold change in the reviewer's comments refers to global trend estimates. These would generally be expected to be much smaller in magnitude than any individual branch changes because they capture the average size of extant species relative to their common ancestor.

The authors choice to study terminal branches rather than all branches is justified by the fact that they do not infer ancestral SSD. However, ancestral states can readily be derived from the output of the fabric model. Regardless, studying terminal branches is an unusual approach to studying evolutionary change – it is ignoring the millions of years of shared evolutionary change among species and only looking at the last few million years or so of independent evolution. In fact, looking at only terminal branches could be very misleading as the authors themselves acknowledge (L114-115) that SSD is not randomly distributed across the phylogenetic tree. Because of this, it is expected that changes along internal branches leading to clades are likely to have profound impacts on the evolutionary distribution of SSD across tetrapods.

We agree that exploring all branches is interesting. We originally focused on analysing the data from shifts at the tips of the tree for two reasons. First, uncertainty in inference of evolutionary change increases from tip to root and focusing on the tips means that we analyse the most robust estimates of evolutionary change. Second, and contrary to the reviewer's assertion, ancestral states from the Fabric model could not (at the time) be readily derived from the output of BayesTraits. Ancestral states were not part of the output from BayesTraits or the post-processing software provided by the same authors. We explored the possibility of inferring ancestral states from the output but could not come up with a satisfactory way to do this (in contrast to pure rate variable models, it is not possible to simply estimate ancestral states by fitting a Brownian model to the rate scaled tree). Therefore, we contacted Dr Andrew Meade, the author/maintainer of the BayesTraits software. Dr Meade confirmed that the Fabric model did not provide ancestral state estimates. Dr Meade kindly offered to code ancestral state estimation into the model fitting step including incorporating the inferred beta estimates. We then re-ran *all* BayesTraits analyses using this new version. This enabled us to infer ancestral states for male and female size and therefore to calculate estimates of SSD at each node in the tree. We now include these ancestral states estimates in our comparisons of the frequency of directional evolution in relation to the mode of SSD (lines 203-226). This expanded analysis does not alter our previous conclusions. Because PGLS models tip data, we retain our analysis of only tips for studying the magnitude of directional evolution.

Note that we offered to add Dr Meade as a co-author for his contribution but he declined.

It comes to my attention here that the authors do not determine the significance of the global beta in the fabric model – while they discuss the global trend models it is absolutely essential to compare the statistical fit of a model with and without global trends – for example, using marginal Lhs. If a global trend is found to be non-significant, all subsequent analyses are rendered meaningless – the betas will be very different between the two models.

As part of our re-analyses we now include comparisons with the global trend model (supplementary table S1). Using marginal likelihoods we found the Fabric + trend was always the best fitting model, albeit sometimes only marginally. We run all downstream analyses on the Fabric + global trend model in each case.

Minor points:

- The null hypothesis (line 342) should specify clearly that SSD would be derived from a non-directional or unbiased random walk – the fabric model with betas is also a random walk model.

Thanks for highlighting this. We have reworked this whole section and clarified that the Fabric model is also a random walk model.

- On line 328, the authors state that the ancestral trait estimates are identified – but they are only estimating the root estimate here, not the trait estimates for all ancestors. This should be clarified (although see above).

In our new analyses ancestral states are estimated for all nodes.

- Did the authors test the effect of priors on their analysis? While they suggest they followed recommendations in Pagel et al (L331-335), the priors recommended in this paper were specifically for log₁₀ mammalian body size measured in kg. Here, the authors are studying natural logged body size of tetrapods measured in grams.

We examined our priors carefully to ensure that they reflect the distribution of body sizes in our data. Our priors are defined on page 14, lines 375-378.

- On the subject of priors, the authors should also specify the reasoning behind the gamma prior on the ancestral trait estimate – what does this distribution centre on in units of body size – and why? Is it based on fossil estimates for the size of the basal tetrapod?

Our priors are centred on the mean value of species per tetrapod class in our data. The priors are sufficiently liberal to allow inference of states outside the range of extant species.

Reviewers' comments:

Reviewer #3 (Remarks to the Author):

Summary

In this paper, the authors attempt to identify some of the causal mechanisms behind the evolution of SSD. They use PCMs and a dataset of tetrapods to estimate directional changes in body size evolution, which are then used to infer differences in size changes between sexes. They suggest several findings, including that directional selection on body size is typical for tetrapods, the patterns of SSD necessitate an explanation of directional evolution, and that selection on size generating SSD is typically stronger in males. Overall, I believe the authors have tackled an interesting problem using a rich dataset. Nonetheless, I have concerns about the methodology employed by the authors and whether some of the claims listed above are substantiated by their results.

Major Comments

My primary concern with this work is the authors' choice to analyze male and female body size separately. I understand this decision allows the exploration of whether shifts in SSD are associated with shifts in male or female body size, but I believe this choice may lead to undesirable consequences. First, male and female body sizes are not independent; they are highly correlated with one another, and SSD is highly correlated with body size in general (Fairbairn 1997). However, in all downstream statistical tests in this manuscript, they seem to be treated as independent variables, such that shifts in male body size are examined and compared independently to shifts in female body size. Given the highly correlated nature of these variables, I would like the authors to explain why this would not result in inflated significance. If the body size of the lineage increases in general, we can expect both male and female body sizes to increase. Modeling shifts independently ignores this fact. One solution could be to model SSD directly. Treating SSD as a single variable (rather than as two independent variables for male and female body sizes) should resolve this problem. Once shifts in SSD are identified, the authors could then examine shifts in male and female body sizes separately to qualitatively assess when shifts are driven by males or females.

The reviewer raises interesting and important questions regarding which traits should be treated as response variables in analyses of SSD. We certainly agree with the reviewer that sex of males and females is strongly correlated, and we mention this explicitly at multiple locations in the text (lines 82-84, 193-195). However, we respectfully disagree with the reviewer that this means our approach of analysing them independently is flawed. Most shifts in size are shared between the two sexes, occur on the same

branches and in the same directions – this is evident when examining the figures in Supporting Information S2, and wholly unsurprising considering the strong correlation between male and female size (lines 193-195). However, we think the differences, when they occur, are extremely telling and important for understanding the evolution of SSD. Therefore, we go into great depth into exploring both difference in regime – *i.e.*, if shifts of one type or other are more common in one sex or the other (section **Directional shifts are more common in males**) – and differences in magnitude – *i.e.*, if changes are larger in one sex or the other (section **Sex-specific differences in the magnitude of directional shifts are correlated with SSD**). The new names of the sections, as well as the reshuffling of some of the text, was done in order to make our intent and reasoning clearer to the reader.

The authors conduct several binary tests. The repeated tests will lead to inflated type I error, and some form of correction (e.g., Bonferroni) should be applied. Additionally, the section beginning at Line 202 should report effect sizes in addition to p-values.

We have now performed a Benjamini-Hochberg false-discovery-rate correction to *p-values* and report them these values in the text.

One of the conclusions the authors make is that SSD in tetrapods "cannot be explained without directional evolution." This conclusion is based on their modeling results using Fabric, which compares directional selection against a model of shifting variance and a more standard Brownian motion model. I do not believe this is a comprehensive enough model set to make that claim. Two potential alternatives are a model of stabilizing selection and a model where trait disparity has decreased over time. These are the Ornstein-Uhlenbeck and early burst models from Hansen (1997) and Harmon et al. (2010). These are standard phylogenetic comparative models that have been used to discuss body size evolution. Without including these more standard models, particularly an OU-type process, it is difficult to trust that directional selection is necessary for SSD. I acknowledge that it may be challenging to directly compare the Fabric model to an OU process, but an OU model should result in a variance pattern similar to the directional model. Perhaps a simulation assessment, as the authors did later in the paper, would be appropriate.

We have conducted additional analyses fitting a single stationary peak OU model (consistent with Harmon et al. 2010) and Pagel's delta model. Delta is a model that allows accelerating or decelerating evolution. The latter is analogous to the early burst model. We choose delta over EB for the pragmatic reason that it is implemented in BayesTraits under the same modelling framework and therefore (along with OU) can be compared to the other models by estimating marginal likelihoods using stepping stone sampling. We found no support for either OU or EB (in all cases their fit was hundreds of log-likelihood units lower than Fabric models; see Table S1). These simple alternatives

are also rejected relative to the evolvability model that allows the rate of evolution to vary, but in the absence of direction shifts. The key distinction then is that the Fabric model, that includes both evolvability and directional trends, is a much better fit to the data than any model that excludes directional evolution.

I believe the authors are too liberal in their use of the term "selection." The models they are fitting, like all PCMs, are phenomenological. They describe processes not based on first principles but on potential explanations of patterns. For example, while Brownian motion has a parameter, one would not interpret sigma as the rate of genetic drift. This is because drift, like directional selection, acts on fundamentally different time scales. Thus, while the authors repeatedly make claims about sex-specific selection, I do not believe this is accurate. This language should be considered more carefully. The authors use more appropriate terminology elsewhere in the paper when they discuss directional shifts rather than directional selection.

We thank the reviewer for this insightful comment. We recognise that in the previous version of the manuscript we have perhaps been too liberal with the use of the term 'selection'. While we still stand by the interpretation of directional changes as indicative of selection (and we make clear this is our interpretation in lines 268-270), the reviewer is correct that there could be other processes generating the same patterns, and we have indeed at some places unintentionally muddled the lines between the patterns we observe and the interpretations we infer. We have carefully revised the language throughout the manuscript, making clear divisions between discussions of the directional changes we estimate from our models, and the evolutionary mechanisms such as selection that their presence suggests.

Minor Comments

The claims made from lines 186 to 201 suggest that directional selection tends to result in species that are monomorphic. I was missing some information about the parameter values and what they represent. A directional trend is interesting, but what exactly is being measured? What is the magnitude of the effect in numerical terms?

We have removed this paragraph from the manuscript, since we have followed the editor's recommendation to omit the simulations from the study (see below).

In the section "Selection tends to act more frequently on males," I think it would be helpful to explain what is meant by "within and between sexes." In the methods, you include "e.g., are there more positive than negative changes in male body size in male-biased species?" This sort of explanation could be added when presenting the results.

We have rephrased the entire section, now retitled Directional shifts are more common in males. We have added a short introductory paragraph to the analyses to explain what we aim to test with the binomial tests (lines 196-198), as well as some sentences at the beginning of the discussion of each set of tests to explain more in detail what each one tests (lines 199-202 & 208-211).

Figure 1B: It may be worth adding a graphic that indicates whether values on the x-axis are male- or female-biased. For example, use traditional male and female symbols where, on the rightmost side of the graph (where females > males), the symbol for females is larger than that for males, and vice versa on the leftmost side of the figure. The meaning of the values is also not mentioned in the figure caption.

Done.

Line 98: The brief model description could be worded more clearly. When I first read the sentence, I wasn't sure whether the Fabric model distinguishes between directional changes and evolvability, or if it distinguishes between increases and decreases in both directional changes and evolvability.

We have rewritten this paragraph to increase clarity in the model description (lines 98-100).

Line 232: Does “decreasing male size” mean that as male size decreased, X happened, or does it mean that as the magnitude of negative male size shifts increased, X happened? Based on Figure 3, I would infer the latter, but the text suggests the former. My phrasing “magnitude of negative male size shifts increased” is unclear, so perhaps the authors could clarify this.

We agree that the previous phrasing was confusing. We have rephrased this to read “greater decreases in male size” (lines 222-225).

Line 279: Some of the material in the conclusion would be better placed in the results and discussion. Some new information is presented (e.g., the paragraph starting at line 285), and I think this could form its own section in the discussion, keeping the conclusion more concise.

We have created a new section titled **Inferring mechanisms of SSD evolution**, which we have placed before the **Conclusions**. We have collated into this section some text from the previous section (now renamed to **Sex-specific differences in the magnitude of directional shifts are correlated with SSD**), as well as the paragraphs mentioned here by the reviewer. The **Conclusions** section now includes only a single summary

paragraph.

Line 408: What is meant by “log units”? Is this the log of the parameter value? What are the units of the parameter value?

We have added a clarification both here (line 408) and in the main text (line 219).

Line 421: Be clear that the response variable of directional change is continuous. The phrasing initially led me to think it was categorical.

We have rephrased this sentence accordingly (lines 420-424).

Comments on responses to Reviewer #2:

Point 1: The choice to examine number and direction of shifts was based on point 1 of reviewer 2. In response, the authors changed their approach to look at direction and number of shifts. I do not believe this choice addressed the lack of clarity indicated by the reviewer. Their choice of verbiage comparing within and between sexes as well as within and between males/ females was still confusing even if their new approach made it easier to distinguish between hypotheses.

As described in the responses to the points above, we have rewritten large sections relating to the binomial tests to improve the clarity. We hope our intentions and aims with these tests are now clear.

Point 2: the authors additional analysis and changes to the text satisfactorily addressed the concerns of the reviewer.

Thank you!

Point 3: I agree with the reviewer that the simulations are not necessary for this manuscript. I also do not believe the parameter estimates from the model are sufficiently discussed in the text despite the new table. I also agree with the reviewers comments about the OU model being more appropriate to examine “selection” (though I strongly feel this terminology is not appropriate). The simulations do not provide sufficient support to overcome this critique in my opinion.

We have now omitted the simulations entirely from the manuscript and, as noted above, carefully rephrased throughout to be clear when we are referring to directional evolution (patterns estimated as output from the Fabric model) and selection (our interpretation of those patterns). We also include a simple version of the OU model in

an expanded set of models. The OU model is always rejected in favour of the Fabric model.

Point 4: This was addressed sufficiently.

Point 5: The authors have addressed reviewer 2s concerns.

Point 6: This was addressed by the authors.

Minor points: Sufficiently addressed by the authors.

Thank you!